# Capturing nascent extracellular vesicles by metabolic glycan labeling-assisted microfluidics

Qiuyue Wu[1], Wencheng Wang[1], Chi Zhang[1], Zhenlong You[1], Yinyan Zeng[1], Yinzhu Lu[1], Suhui Zhang[1], Xingrui Li[1], Chaoyong Yang [1,2] & Yanling Song [1] ✉

Extracellular vesicle (EV) secretion is a dynamic process crucial to cellular communication. Temporally sorting EVs, i.e., separating the newly-produced ones from the pre-existing, can allow not only deep understanding of EV dynamics, but also the discovery of potential EV biomarkers that are related to disease progression or responsible to drug intervention. However, the high similarity between the nascent and pre-existing EVs makes temporal separation extremely challenging. Here, by co-translational introduction of azido groups to act as a timestamp for click chemistry labelling, we develop a microfluidic-based strategy to enable selective isolation of nascent EVs stimulated by an external cue. In two mouse models of anti-PD-L1 immunotherapy, we demonstrate the strategy's feasibility and reveal the high positive correlation of nascent PD-L1$^+$ EV level to tumor volume, suggesting an important role of nascent EVs in response to immunotherapy in cancer treatment.

EVs are nano-sized vesicles secreted by virtually all cells[1]. Their role in mediating cell-cell communication is known to be crucial in a plethora of physiological and pathological processes, making them potential biomarkers for many diseases, such as cancer, neuro-degenerative diseases, and virus infection[2–4]. Depending on the biological context, EV secretion is variable. Thus, capturing EV dynamics, including selectively sorting nascent EVs from the total, will inform us of the essential biological response to specific stimuli, which can not only deepen our understanding of EV regulatory mechanisms but also lead to the discovery of potential EV biomarkers for disease progression and treatment response[5]. Classical EV-separation methods, including sedimentation rate-based ultracentrifugation[6], size-based microfiltration[7], and affinity-based magnetic bead/microfluidic enrichment[8–11], are mostly based on the inherent physicochemical parameters[12] of EV that lack of temporal resolution, thus incapable of distinguishing nascent EVs from the pre-existing. Time-lapse has been harnessed to study EV dynamics in cultured cells[13], however, it is still limited by the inability to reflect complex EV exchanging under physiological conditions, which is critical for understanding EV-mediated intercellular communication in vivo. Consequently, how the production rate and molecular alteration of nascent EVs affect their functional outcomes as well as the metabolism of recipient cells remains an important question yet to be answered[5]. The challenge mainly resides in the presence of the vast majority of pre-existing EVs and the dynamic nature of de novo production of nascent EVs.

Here, we develop a microfluidic-assisted enrichment strategy based on metabolic labeling and click chemistry (Melac-Chip) to allow the selective isolation of nascent EVs stimulated by an external cue (by immunotherapy in our case, Fig. 1). During anti-PD-L1 immunotherapy, we conducted in vivo metabolic glycoengineering with unnatural sugars to ubiquitously label the newly-produced EVs with azido groups (in analogy to mark letters with timestamps) to distinguish them from the pre-existing populations. Subsequent click chemistry linked the azido-labeled EVs with alkynyl biotin[14], allowing them to be specifically

---

[1]State Key Laboratory of Physical Chemistry of Solid Surfaces, Key Laboratory for Chemical Biology of Fujian Province, The MOE Key Laboratory of Spectrochemical Analysis & Instrumentation, Department of Chemical Biology, College of Chemistry and Chemical Engineering, Xiamen University, Xiamen 361005, P. R. China. [2]Institute of Molecular Medicine, Renji Hospital, Shanghai Jiao Tong University School of Medicine, Shanghai 200127, China. ✉e-mail: ylsong@xmu.edu.cn

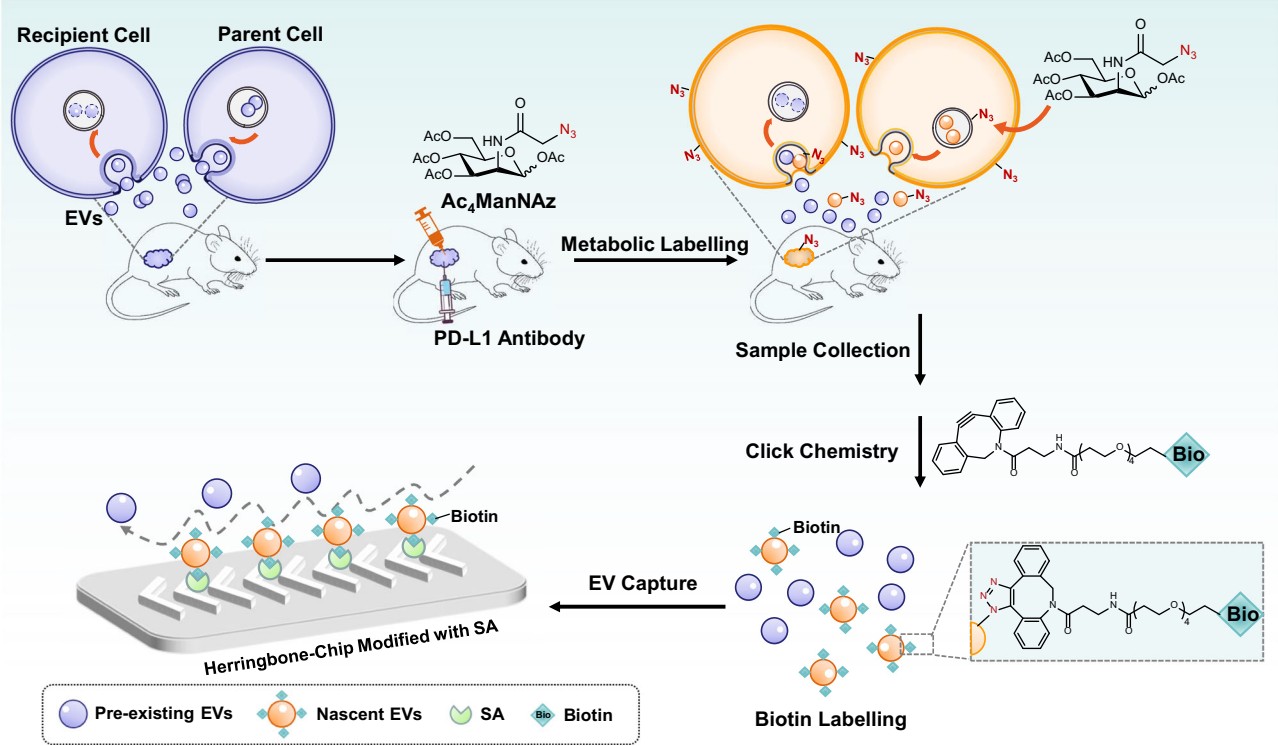

**Fig. 1 | Schematic representation of Melac-Chip to capture the in vivo dynamics of newly-produced EVs during immunotherapy.** Tumor-bearing mice were simultaneously treated with PD-L1 antibody and Ac₄ManNAz (tetraacetylated N-Azidoacetyl-mannosamine), an unnatural sugar for metabolic labeling of EVs with azido groups. Subsequently, the azido-labeled EVs were tagged with biotin groups by click chemistry and captured by streptavidin (SA)-modified herringbone microfluidic chip.

and efficiently captured by a streptavidin-modified herringbone chip, which can generate steady chaotic flows to enhance the collisions between EVs and the affinity interface. With the Melac-Chip strategy, we are able to quantify the relative amount of EV produced after each immunotherapy using sequential administration.

## Results and discussion

### Testing Metabolic Glycan Labeling (MGL) in vitro

An ideal EV timestamp ought to be biorthogonal and biocompatible with EV functioning[14]. Taking that into consideration, we chose Ac₄ManNAz, an unnatural azido-containing monosaccharide, as our MGL additive since it has been widely exploited for membrane glycoprotein labeling ex vivo and in vivo with high efficiency and biocompatibility[15–17]. Meanwhile, we selected EV isolated from cell culture media as the model vesicle system to test whether the azido groups can be efficiently incorporated into the vesicle through the labeled membrane glycoproteins. After treating human melanoma A375 cells with Ac₄ManNAz, we collected the secreted A375 EVs by ultracentrifugation[18–20], loaded them on latex beads[21], and then stained them with alkyl dye (DBCO-Cy5) capable of reacting with azido groups[22] (Fig. 2a). Flow cytometry analysis (Fig. 2b) and confocal imaging (Supplementary Fig. 1) displayed apparent fluorescence signals from the dye on the latex beads containing MGL EVs only (in comparison with the non-MGL control), suggesting the successful incorporation of azido groups. The detected fluorescence intensity reached the plateau at the conditions of 50 μM Ac₄ManNAz (60-h incubation) and 12.5 μM DBCO-Cy5 (1-h incubation).

To assess the biocompatibility of Ac₄ManNAz on EVs, we compared their physical properties and protein contents with and without MGL. Transmission electron microscopy (TEM) showed the cup-shaped structure for the MGL EVs (Fig. 2c), the mean diameter and zeta potential of which were measured to be 140 nm and -4.5 mV,

respectively, by dynamic light scattering (Fig. 2d, e). All of these properties look alike to those from the non-MGL EVs. In addition, the labeled EVs showed high expression of CD63 (Fig. 2f), a classical pan-EV marker[4]. Further proteomics study revealed that out of 1,000 proteins, a total of 806 were shared by the MGL and non-MGL EVs (Fig. 2g). Gene ontology analysis of these proteins and their annotated biological processes (Fig. 2h), cellular components (Fig. 2i), and molecular functions (Fig. 2j), suggested highly similar protein contents for the EVs with and without MGL. Moreover, using the top 30 proteins as a reference, we confirmed the great reproducibility of the MGL on EV generation (a correlation of ~0.99 among three biological replicates, Fig. 2k, l). Based on all of these results, it is reasonable to conclude that Ac₄ManNAz additive is biocompatible with EV production and affects little on EV morphology and proteome.

### Microfluidic isolation of EVs with in vitro MGL

To isolate the MGL EVs, a biorthogonal reaction was performed by using alkyl biotin (12.5 μM DBCO-PEG₄-biotin for 1 h) to click with the EV azido group, the products of which were later captured by streptavidin (SA) modified microchip (Fig. 3a) with an optimized flow rate of 1.25 mL h⁻¹ (Supplementary Fig. 2) (unless otherwise specified, these conditions were used in the subsequent analysis by Melac-Chip). Due to the lack of azido groups, those non-MGL EVs were unable to link with the biotin modification, thus being washed away during the microchip capturing. To promote liquid-solid mass transfer, we chose herringbone microchip (HB-Chip) as the high-throughput microfluidic mixing device. Consistent with the previous work[23,24], by simulation the herringbone patterned channels were able to disrupt the laminar flow profile and generate steady chaotic flows (Fig. 3b), which can enhance the collisions between EVs and the affinity interface. After capturing, the detection signal was read out and amplified by the fluorescence enzyme immunoassay via sequential introduction of anti-CD63, β-

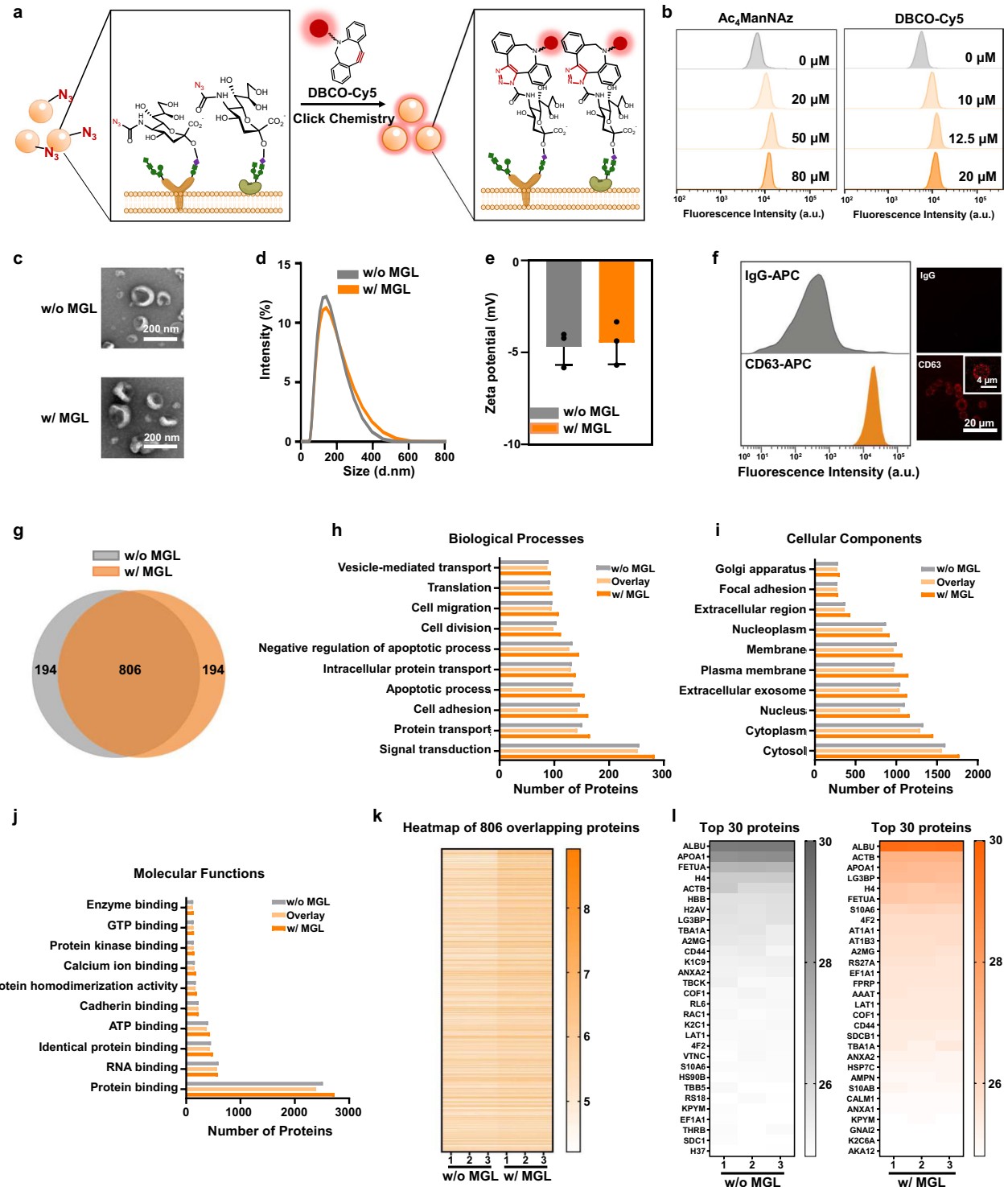

**Fig. 2 | Characterization of the azido-labeled A375 EVs. a** Schematic diagram of the click reaction between the azido-modified A375 EVs and the alkyl dye DBCO-Cy5. **b** Fluorescence intensity of the A375 EVs treated with different concentrations of Ac₄ManNAz (left) or DBCO-Cy5 (right). The EVs were loaded on latex beads, and then reacted with the DBCO-Cy5 dye prior to characterization. **c** Morphology comparison of the MGL A375 EVs and non-MGL A375 EVs by TEM. Scale bar = 200 nm. **d, e** Comparison of the size distribution (**d**) and Zeta potential (**e**) of the MGL (w/ MGL) A375 EVs and non-MGL (w/o MGL) A375 EVs by dynamic light scattering. $n = 3$ biologically independent experiments. Data shown as mean ± SD. **f** Characterization of CD63 expression on the azido-labeled A375 EVs. **g** Veen diagram to compare the A375 EV proteins before and after MGL. **h–j** Biological processes (**h**), cellular components (**i**) and molecular functions (**j**) of gene ontology (GO) enrichment analysis of proteins detected in A375 EVs with (in orange) or without (in gray) MGL. Proteins co-expressed in both are indicated in light orange. **k** Heatmap of the expression levels of the 806 proteins detected both in azido-labeled or unlabeled A375 EVs. The log 10 expression values for the overlapping proteins are indicated by the colors shown in the scale. **l** Heatmap with the top 30 proteins (in triplicate) of EVs from A375 cells with (in orange) or without (in gray) Ac₄ManNAz treatment. The expression changes are indicated in log 2 scale. Source data are provided as a Source Data file.

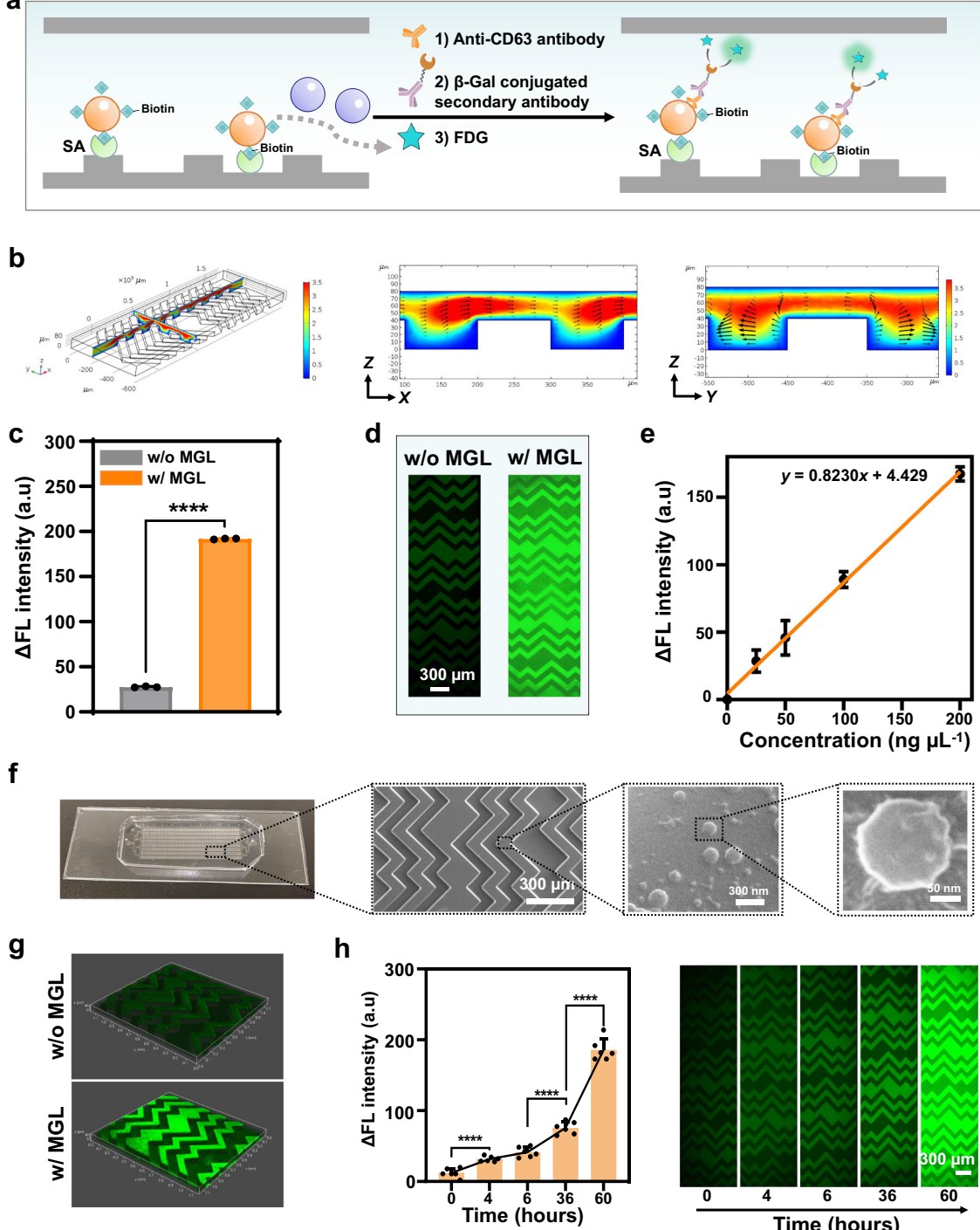

**Fig. 3 | Microfluidic isolation of EVs with in vitro MGL. a** Schematic illustration of biotin-linked MGL EVs captured by a streptavidin modified chip and detected by a fluorescence enzyme immunoassay. **b** Numerical simulations of velocity streamlines and turbulence flows formed on microgrooves in Z-X and Z-Y vertical cross sections of the herringbone chip channel. **c** The fluorescence intensity of MGL A375 EVs (in orange) and non-MGL A375 EVs (in gray) captured by Melac-chip. ΔFL = FL − $FL_0$, where $FL_0$ and FL are the fluorescence intensity detected by Melac-Chip before and after the addition of EVs. Statistical significance was determined by a two-tailed unpaired $t$-test. $P = 6.8479 \times 10^{-10}$, ****$P < 0.0001$. $n = 3$ biologically independent experiments. Data shown as mean ± SD. **d** Representative fluorescence images of MGL A375 EVs and non-MGL A375 EVs captured by Melac-Chip. **e** Calibration curve for quantifying MGL EVs by Melac-Chip. $n = 4$ biologically independent

experiments. Data shown as mean ± SD. **f** Scanning electron microscope (SEM) imaging of herringbone chip to show the surface morphology of the captured MGL EVs. **g** 3D confocal fluorescence microscopy showing the DiI-stained MGL EVs selectively captured by Melac-Chip. **h** Time curve for quantifying MGL EVs by Melac-Chip (left) and the corresponding representative fluorescence images (right). A375 cells were treated with 50 μM Ac$_4$ManNAz for different time (0, 4, 6, 36 and 60 h), then labeled with DBCO-PEG$_4$-biotin (12.5 μM) for 1 h, and finally captured by SA-Chip. $n = 6$ biologically independent experiments. Data shown as mean ± SD. Statistical significance was determined using Tukey's Method with One-Way ANOVA. 0 h vs 4 h, $P = 6.9914 \times 10^{-5}$. 6 h vs 36 h, $P = 2.1714 \times 10^{-5}$. 36 h vs 60 h, $P = 3.1468 \times 10^{-8}$. ****$P < 0.0001$. Source data are provided as a Source Data file.

galactosidase (β-Gal) conjugated secondary antibody and fluorescein di-β-D-galactopyranoside (FDG)[9] (Fig. 3a).

Using the fully developed Melac-Chip strategy, we quantified an increase of ~6.9 folds in fluorescence intensity for the MGL over the non-MGL EVs, after removal of the EV-null background signal (Fig. 3c, d; Supplementary Fig. 3). The observed fluorescence intensity displayed a positively linear relationship with the MGL EV concentration ($R^2 = 0.98$; limit of detection, $LOD = 9.95$ ng μL$^{-1}$), suggesting the quantitative capability of Melac-Chip (Fig. 3e). In addition, EVs captured by Melac-Chip maintained a classical spherical topology (Fig. 3f), implying their well integrity. Moreover, 3D reconstructed fluorescence images indicated that only the MGL EVs (pre-stained) were captured (Fig. 3g), further highlighting the excellent selectivity of Melac-Chip. We also explored the time window of Melac-Chip by using it to analyze the A375 EVs collected at different time points (0-60 h) after MGL. The results showed an increased fluorescence signal over time and MGL EVs can be detected as early as in 4 h (Fig. 3h), suggesting the timeliness of MGL and high efficiency of Melac-Chip.

## Microfluidic isolation and detection of EVs with in vivo MGL

We next explored the metabolic labeling of EVs in vivo and the corresponding EV isolation and detection. We chose 4T1 breast tumor-bearing mice as the model system and injected Ac₄ManNAz (20 μL of 40 mM Ac₄ManNAz in intratumoral injection and 100 μL of 140 mM

Ac₄ManNAz in intraperitoneal injection) into the mice once daily for three consecutive days[25,26] (PBS injection as the control, Fig. 4a). To monitor biodistribution of the azido-modified EVs, we collected the mouse samples of plasma, tumor tissue, and tissue from four major organs (heart, liver, lung and kidney). EVs in the plasma samples were directly analyzed by Melac-Chip, while EVs from tissue samples were separated by using a combination of slicing and enzyme digestion followed by differential ultracentrifugation[27] (Fig. 4b), and then subjected to Melac-Chip analysis. To enable the analysis, only 5 μL of each sample was required to click with DBCO-PEG₄-biotin, and the analysis showed a similar phenomenon to that of the in vitro MGL, that is, the azido-labeled samples (regardless of from plasma or tissue) displayed apparent increases in the intensity of anti-CD63 staining as comparing to the PBS control (Fig. 4c), suggesting the successful MGL on EVs in vivo and the isolation capability of Melac-Chip on the in vivo labeled EV samples. Among the quantified EV levels from the in vivo MGL model, those from the plasma and tumor tissue samples were the highest (~43-fold higher to the lowest) (Fig. 4c), indicating that during the 3-day in vivo MGL window, the labeled EVs were primarily distributed within the tumor and blood. Considering the high EV level and easiness to handle, we chose mouse plasma samples for subsequent experiments.

As the tumor-specific antibodies are integrated into Melac-Chip for EV detection, in principle the strategy could be applied for analysis

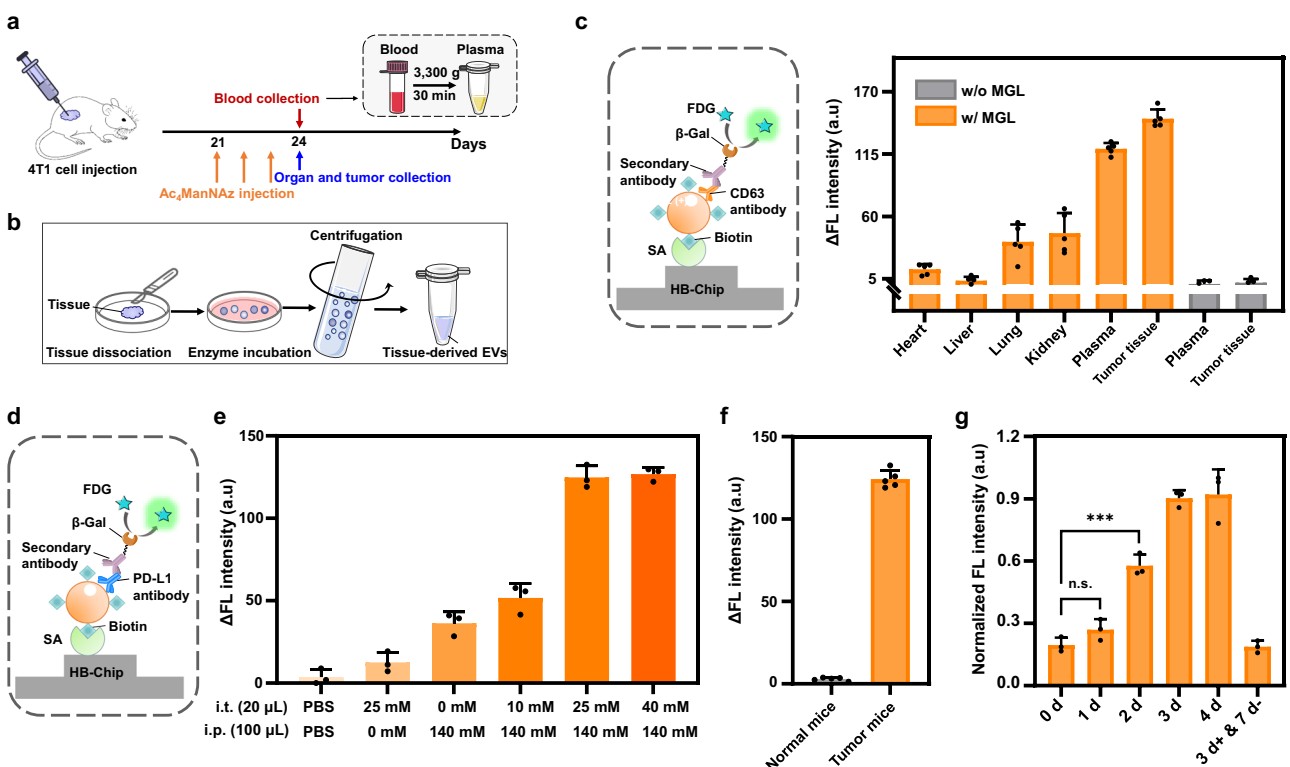

**Fig. 4 | Isolation and detection of EVs with in vivo MGL. a** Schematic schedule of the in vivo MGL in a 4T1 tumor-bearing mouse model. **b** Protocol diagram of the collection process for tissue-derived EVs. **c** Schematic of the detection of biotin-linked MGL CD63⁺ EVs and the detected distribution in plasma and different tissue samples. The fluorescence detection was on CD63⁺ EVs from plasma, tumor tissue and other major organs of 4T1 tumor-bearing mice with MGL, as well as CD63⁺ EVs from plasma and tumor tissue of 4T1 tumor-bearing mice without MGL. ΔFL = FL − FL₀, where FL₀ and FL are the fluorescence intensity detected by Melac-Chip before and after the addition of sample. *n* = 5 biologically independent experiments. Data shown as mean ± SD. **d** Schematic of the detection of biotin-linked MGL PD-L1⁺ EVs. **e** The detected fluorescence intensity of PD-L1⁺ EVs from plasma samples of 4T1 tumor-bearing mice (*n* = 3 biologically independent experiments) with different

injection manners of Ac₄ManNAz. I. t. refers to intratumoral injection and i. p. refers to intraperitoneal injection. Data shown as mean ± SD. **f** The detected fluorescence intensity of PD-L1⁺ EVs from plasma samples of normal mice and 4T1 tumor-bearing mice with Ac₄ManNAz treatment (*n* = 5 biologically independent experiments). Data are presented as mean values ± SD. **g** Normalized PD-L1 fluorescence intensity of plasma samples (*n* = 3 biologically independent experiments) from 4T1 tumor-bearing mice at 0, 1, 2, 3 and 4 d injection of Ac₄ManNAz (once/day). "3 d + and 7 d -" indicates a stop for 7 days after 3 consecutive days of injection. Data shown as mean ± SD. Statistical significance was determined using Tukey's Method with One-Way ANOVA. *P* = 0.1063 (0 d vs 1 d), and *P* = 0.0005 (0 d vs 2 d). ****P* < 0.001, and n.s. indicates non-significance (*P* > 0.05). Source data are provided as a Source Data file.

of tumor-derived EVs, even though the Ac$_4$ManNAz-based MGL is not tumor-selective. To demonstrate this speculation, we tried to harness the anti-PD-L1 antibody to detect the PD-L1 positive (PD-L1$^+$) EVs (Fig. 4d), a promising marker for tumor diagnosis[20,28,29], from the total captured ones. On the basis of confirming the PD-L1 expression on 4T1 cells (Supplementary Fig. 4), we firstly investigated how the administration manner affects the efficiency of EV labeling in 4T1 tumor-bearing mouse model. Considering both intratumoral (i. t.)[26,30] and intraperitoneal (i. p.) injection[25,31] are commonly used for Ac$_4$ManNAz-based cell labeling in vivo, we gave a series of i. t. and i. p. injection to 4T1 tumor-bearing mice (once daily for three consecutive days). Among them, using Melac-Chip we identified the optimal condition to be the combination of 20 μL of 25 mM Ac$_4$ManNAz in i. t. and 100 μL of 140 mM Ac$_4$ManNAz in i. p., under which the anti-PD-L1 antibody staining intensity reached the highest, ~34-fold to that of the control group (PBS injection, Fig. 4e). Meanwhile, although PD-L1 is known to be expressed by immune cells in the plasma[32], at this condition nearly negligible PD-L1$^+$ MGL EV signal was detected in the plasma samples from normal mice (Fig. 4f), suggesting low expression of PD-L1 on EVs in normal plasma samples[20,33,34] and supporting the tumor-biomarker role of PD-L1$^+$ EVs[20,29]. We also studied the time window of MGL for efficient detection of these EVs with Melac-Chip. Again, we administered Ac$_4$ManNAz (20 μL of 25 mM Ac$_4$ManNAz in i. t. and 100 μL of 140 mM Ac$_4$ManNAz in i. p.) once daily to 4T1 tumor-bearing mice for up to 4 days. We collected the mouse plasma samples each day starting from day 0. Melac-Chip analysis revealed the detectable signal of PD-L1$^+$ MGL EVs on day 2 and an increase in the signal till day 3 when it seemed to be saturated (Fig. 4g). As the detected PD-L1$^+$ MGL EV signal reached the plateau on day 3, a stop of Ac$_4$ManNAz injection for consecutive 7 days (referring to the classical interval between two immunotherapy doses[35–37]) completely diminished the signal of this EV, suggesting the well clearance of this unnatural sugar in vivo and the well sensitivity of Melac-Chip. Together, these results demonstrate that Melac-Chip can be used to not only isolate the in vivo metabolically labeled EVs but also selectively detect tumor-derived EVs from the total isolation.

**Probing nascent EVs in response to anti-PD-L1 immunotherapy**
Immune-checkpoint inhibitors targeting PD-L1/PD-1 axis have shown remarkable promise in tumor treatment. However, only 10-40% of patients can derive benefit from this immunotherapy[37–39]. Thus, to help the patients choose the right treatment and to improve the treatment efficacy, it is important to be capable of predicting patient response from the therapy. Currently, increasing evidence points to EV PD-L1 as a predictor for anti-PD-L1/anti-PD-1 therapy, since it can regulate tumor progression through inhibition of T cell activity[20,28,29,40]. Here, using Melac-Chip we are able to selectively probe nascent EVs and reveal their timely response to anti-PD-L1 immunotherapy.

We firstly established an immunotherapy response model on 4T1-bearing mouse[36] by annotating PD-L1 antibody (PBS injection as the control) on day 7 and once/week thereafter for 4T1 breast tumor mouse (Fig. 5a). Simultaneously, we weekly injected Ac$_4$ManNAz with the immunotherapy for three consecutive days to ensure the saturation of in vivo MGL on nascent EVs according to the results of our in vivo MGL test (Fig. 4g). During this process, we observed the suppression on tumor growth in the anti-PD-L1 antibody treatment group compared to the untreatment group (Fig. 5b, Supplementary Fig. 5). The results of hematoxylin-eosin (HE) and immunofluorescence staining further demonstrated the effectiveness of anti-PD-L1 immunotherapy as increased apoptotic cells (anti-TUNEL staining) and infiltration of CD8$^+$ T cells, as well as decreased cancer cells (anti-Ki67 staining) were shown (Fig. 5c).

We then collected and analyzed the mouse plasma samples at a series of time points post injection by Melac-Chip (Fig. 5a). After calibration (Supplementary Fig. 6a), we calculated the concentrations of

the newly-produced CD63$^+$ EVs and found no significant difference between the treated and untreated groups (Fig. 5d), suggesting that the total plasma CD63$^+$ EV production remains stable regardless of tumor growth and drug stimulation. Next, we moved to the nascent PD-L1$^+$ EVs. Since the core of Melac-Chip is click chemistry-based capture followed by primary antibody recognition and secondary antibody amplification, the possible presence of anti-PD-L1 coverage, that is, certain PD-L1$^+$ EVs may carry anti-PD-L1 antibody arising from the therapy, would in principle not affect the capturing (dependent on the azido-modification) nor the amplification (similar secondary-antibody-recognizable Fc regions on both the therapeutic and the detection anti-PD-L1 antibodies). To demonstrate this speculation, we extracted MGL EVs from 4T1 cells and used them as a model to test the detection by Melac-Chip in the presence or absence of pre-incubation with the therapeutic anti-PD-L1 antibody, which showed a detectable PD-L1$^+$ EV level close to each other (Supplementary Fig. 7). Similar to the data processing of nascent CD63$^+$ EVs, we established a calibration curve for the PD-L1$^+$ EVs (Supplementary Fig. 6b) and then estimated that the treatment had a significant impact on these EVs' production (Fig. 5e), which positively correlated to the tumor volume, with a slight decrease from day 10 to 17 and a slight increase in the week after (Fig. 5b, e). As a comparison, instead of Melac-Chip we tried a regular Chip-based method (CD63 antibody for capture) that allows for the quantification of total PD-L1$^+$ CD63$^+$ EVs (a predictor for immunotherapy response[20]) (Supplementary Fig. 6c, Fig. 5f). It turned out that the total PD-L1$^+$ CD63$^+$ EVs showed a good correlation with the tumor volume only in the untreated group. In the treated group, levels of these EVs increased till day 10 and then stayed stable thereafter, which was poorly correlated with the change trend of the tumor volume during this time period (Fig. 5b). Further Pearson correlation analysis confirmed the good correlation between nascent PD-L1$^+$ EVs and tumor volume in the treated ($R^2 = 0.9611$) and untreated ($R^2 = 0.9958$) groups (measured by Melac-Chip), as well as the good correlation between the total PD-L1$^+$ CD63$^+$ EVs and tumor volume in the untreated ($R^2 = 0.8886$) but not the treated ($R^2 = 0.2663$) group (measured by a Chip-based method) (Fig. 5g). These data suggest that nascent PD-L1$^+$ EVs that are only measurable by Melac-Chip can serve as a more universal indicator for tumor progression and immunotherapy response than total PD-L1$^+$ CD63$^+$ EVs, a previously used indicator[20].

Besides, we were also curious about the change trends of the nascent and pre-existing PD-L1$^+$ CD63$^+$ EVs. After using the azido-induced isolation to remove the nascent EVs instead of detecting them, we were able to capture the pre-existing CD63$^+$ EVs by anti-CD63 antibody and then detect the PD-L1 level by anti-PD-L1 antibody (See Methods "Microfluidic detection of EVs with in vivo MGL" for details). The levels of nascent PD-L1$^+$ CD63$^+$ EVs were generated via the subtraction of the pre-existing ones from the total that were measured in Fig. 5f. By plotting these data together (Supplementary Fig. 8), we found that in the untreated group both the nascent ($R^2 = 0.8914$) and pre-existing ($R^2 = 0.8547$) PD-L1$^+$ CD63$^+$ EVs exhibited the relatively good correlation to tumor volume, but in the treated group neither of them correlated well ($R^2 = 0.2424$ & $0.1963$ for the former and latter, respectively). Comparing to the high correlation between nascent PD-L1$^+$ EV and tumor volume, the poor correlation between nascent PD-L1$^+$ CD63$^+$ EVs and tumor volume hints that certain subtypes of EVs other than CD63$^+$ EVs are more directly responsible to the immunotherapy.

To show the generality of Melac-Chip, we switched to the B16F10 melanoma-bearing mouse model which is resistant to the immune-checkpoint blockade[41] (Supplementary Figs. 9, 11a–c) for the quantification of nascent PD-L1$^+$ EVs. After calibration (Supplementary Fig. 10), the detected levels of nascent PD-L1$^+$ EVs again exhibited a good correlation to tumor volume in both the untreated and treated group ($R^2 = 0.9741$ vs 0.9902, Supplementary Fig. 11d, g). Meanwhile, regular chip analysis also indicated the good correlation between pre-existing PD-L1$^+$ CD63$^+$ EVs and tumor volume ($R^2 = 0.9528$ vs 0.9586) as well as

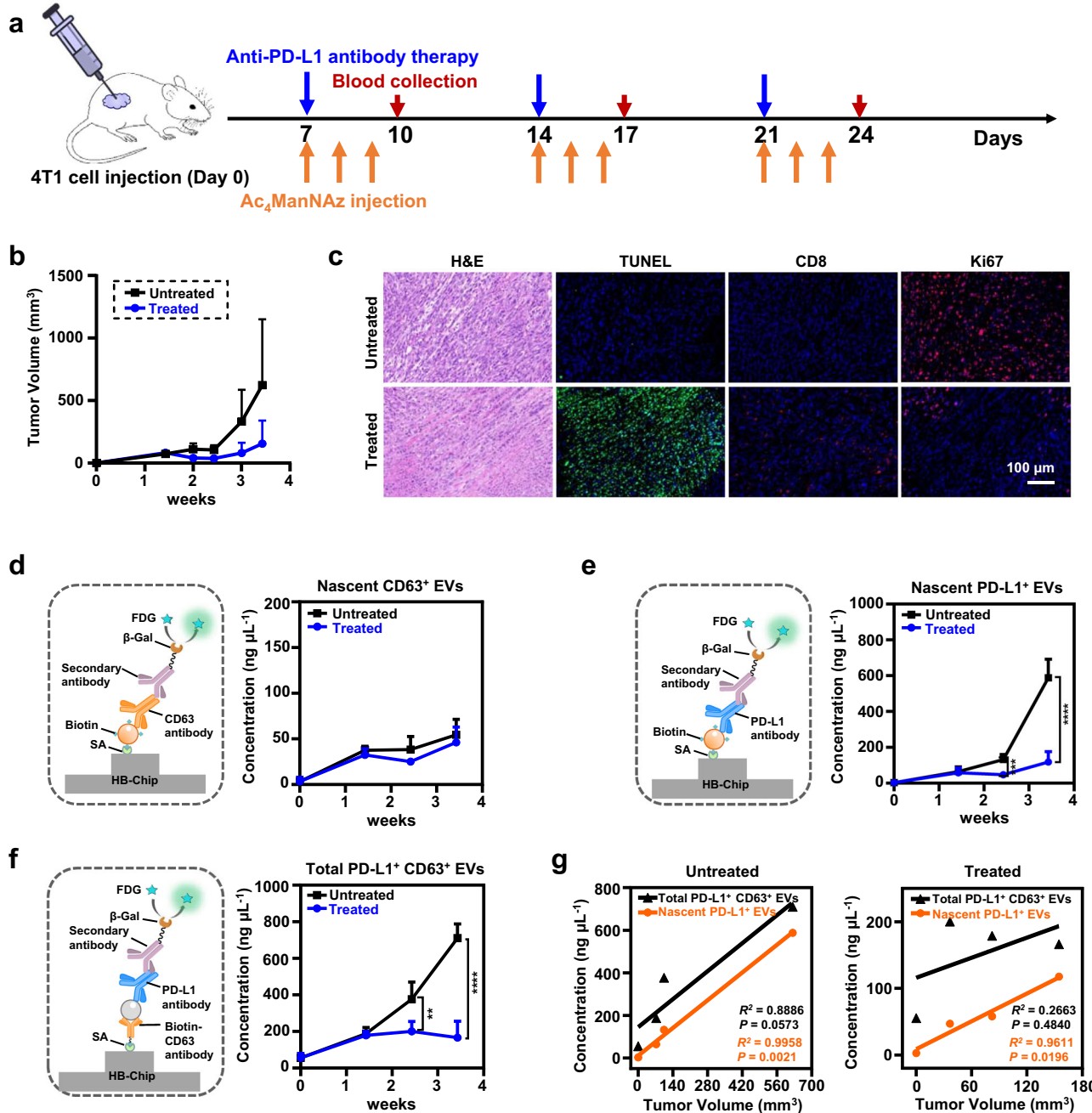

**Fig. 5 | Analyzing nascent EVs in response to anti-PD-L1 immunotherapy.**
**a** Schematic of the tumor implantation, PD-L1 antibody immunotherapy, metabolic glycan labeling and sample collection on a 4T1-bearing mouse model. **b** Tumor growth curves of 4T1-bearing mice with/without PD-L1 antibody treatment. Data shown as mean ± SD. **c** Representative images of tumor tissue by HE and immunofluorescent staining. **d–f** Schematics of the detection of nascent CD63+ EVs (**d**), nascent PD-L1+ EVs (**e**), and total PD-L1+ CD63+ EVs (**f**) as well as the detected concentrations at different time points. Data shown as mean ± SD. Statistical significance was determined by a two-tailed unpaired *t*-test. For nascent PD-L1+ EVs,

treated vs untreatment group: $P = 0.0001$ (17 d), $P = 5.2209 \times 10^{-6}$ (24 days) (**e**). For total PD-L1+ CD63+ EVs, treated vs untreatment group: $P = 0.0035$ (17 d), $P = 2.0980 \times 10^{-6}$ (24 days) (**f**). \*\*$P < 0.01$, \*\*\*$P < 0.001$, \*\*\*\*$P < 0.0001$. **g** Pearson correlation of the nascent PD-L1+ EVs (orange) and total PD-L1+ CD63+ EVs (black) to the tumor volume in 4T1-bearing mice with /without anti-PD-L1 treatment. $n = 5$ biologically independent experiments for the untreatment group, $n = 6$ biologically independent experiments for the treatment group. Correlations were determined by Pearson's *r* coefficient. A two-tailed value of $P < 0.05$ was considered statistically significant. Source data are provided as a Source Data file.

total PD-L1+ CD63+ EVs and tumor volume ($R^2 = 0.9712$ vs 0.9220) in the two groups (Supplementary Fig. 11e–g). However, among the three indexing vesicles, nascent PD-L1+ EVs still correlated best to tumor volume, regardless of immunotherapy or not, which coincides well with the results in the 4T1 breast cancer mouse model. Therefore, Melac-Chip enables detection of nascent PD-L1+ EVs in both immunotherapy responded and resistant models, providing a timely and accurate reflection of tumor progression and therapy effect.

A thorough understanding of EV function requires a dynamic view. Temporal differentiation is essential if one is to describe the dynamic response of EV to specific stimulus. Elucidating these dynamic changes in EV production at a given state is challenging and requires specialized techniques, which should be biocompatible and sensitive enough to quantify subtle changes in EV levels. Based on those techniques, taking a series of "snapshots" along EV production will help us understand the temporal dynamics and provide invaluable

insights for treatment assessment, progression monitoring, and drug development.

Melac-Chip allows the temporal-selective labeling and antigen-specific detection of the nascent EVs, thereby accurately revealing EV response to an external cue. The strength of the approach lies in that metabolic glycan labeling facilitates the efficient differentiation of nascent EVs against a bewildering background of pre-existing ones. Through subsequent click chemistry and mixing-accelerated herringbone microfluidics, newly-produced EVs can be selectively isolated for specific antibody-based quantification. Here we demonstrate that metabolic glycan labeling of EV is biologically compatible (labeling does not appreciably affect EV morphology and proteome) and efficient (4 h of labeling is sufficient to EV identification on chip). Using Melac-Chip we were able to detect the production of nascent PD-L1$^+$ EVs after each anti-PD-L1 administration, and prove nascent PD-L1$^+$ EVs to be a more accurate predictor than total PD-L1$^+$ CD63$^+$ EVs for tumor progression and therapy response. Overall, the combination of EV metabolic labeling and efficient microfluidic enrichment improve our ability to accurately analyze EV secretion over time, which should enable the study of EV secretion mechanisms as well as the in-depth exploration of EV biological function and clinical value. Besides adding a temporal dimension to our understanding of EV dynamics in immunotherapy, in principle Melac-Chip can also be applied to study EV dynamics that is stimulated by a broad range of cues, such as microbial infection, environmental (temperature, light, and gravity) variation, and diet change.

## Methods

### Ethical statement
This research complies with all relevant ethical regulations. All animal studies were conducted in accordance with the National Institute Guide for the Care and Use of Laboratory Animals. The experimental protocols (XMULAC20220298) were approved by the Institutional Animal Care and Use Committee (IACUC) of Xiamen University. All experimental measurements are provided as Supplementary Data inside an Excel file named "Source Data".

### Materials
Phosphate-buffered saline (PBS) buffer (E607008-0500), glycine and 4% paraformaldehyde fix solution (E672002-0500) were purchased from Sangon Biotech (ShangHai, China). Dulbecco's Modified Eagle Medium (DMEM) and RPMI 1640 medium were obtained from Cytiva (Shanghai, China). Fetal bovine serum (FBS) was purchased from Biological Industries (BI) (Shanghai, China), penicillin streptomycin glutamine was purchased from Gibco (USA). Bicinchoninic acid assay kit was obtained from Epizyme Biomedical Technology Co., Ltd (Shanghai, China). Tetraacetylated N-Azidoacetyl-mannosamine (Ac$_4$ManNAz) was obtained from Shanmu Biological Medicine (Jinan, China). DBCO-PEG$_4$-biotin, collagenase D, streptavidin (SA), 3-Mercaptopropyltri-methoxysilane (MPTS) and dimethyl sulfoxide (DMSO) were purchased from Sigma-Aldrich (USA). Aldehyde/sulfate latex beads, bovine serum albumin (BSA), n-γ-maleimidobutyryl-oxysuccinimide ester (GMBS), SuperBlock™ blocking buffer and anti-human CD63 antibody-APC (10 μL/sample, Cat#A15712) were obtained from Thermo Fisher Scientific Inc. IgG-APC (10 μL/sample, Cat#sc-516612) was purchased from Santa Cruz (Texas, USA). Anti-mouse CD63 antibody (20 μg/mL, Cat#MAB5417) was purchased from R&D systems (USA). Biotinylated anti-mouse CD63 antibody (20 μg/mL, Cat#bs-23032R-Bio) was purchased from Bioss (Beijing, China). FDG was obtained from AAT Bioquest (USA). Mouse anti-human CD63 antibody (20 μg/mL, Cat#556019) was purchased from BD Pharmingen (USA), mouse anti-PD-L1 antibody (20 μg/mL, Cat#NBP1-43262) (detection antibody) was obtained from Novus Biologicals (USA), goat anti-mouse IgG H&L (beta-galactosidase) (120 μg/mL, Cat#ab136775) and rabbit anti-rat IgG H&L (beta-galactosidase) (120 μg/mL,

Cat#ab136716) were purchased from Abcam (USA). 1,1'-dioctadecyl-3,3,3',3'-tetramethylindocarbocyanine perchlorate (DiI) lipophilic dye was obtained from Beyotime Biotech Inc (Shanghai, China). Phosphotungstic acid (P28140-500g) was purchased from Acmec Biochemical Co., Ltd (Shanghai, China). Anti-mouse PD-L1 antibody (12.5 mg/kg, Cat#BE0101) for immunotherapy was obtained from BioXCell (USA). DNase I (AC1711) was purchased from Sparkjade (Shangdong, China). DAPI (G1012) and EDTA K$_2$ Anticoagulation Tube (QX0001) were obtained from Servicebio (Wuhan, China). Streptavidin beads (17511301) was purchased from Cytiva (GE Life) (USA).

### Cell culture
Human melanoma A375 cells (#CRL-1619) were purchased from American Type Culture Collection (ATCC) and cultured in DMEM supplied with 10% (v/v) FBS, 100 μg mL$^{-1}$ penicillin, and 100 μg mL$^{-1}$ streptomycin. Murine breast cancer 4T1 cells (#CL-0007) and B16F10 mouse melanoma cells (#CL-0319) were from Procell Life Science & Technology Co., Ltd. (Wuhan, China), and maintained in RPMI 1640 medium with 10% (v/v) FBS, 100 μg mL$^{-1}$ penicillin, and 100 μg mL$^{-1}$ streptomycin. The cells were grown at 37 °C in humidified atmosphere of 5% CO$_2$. All cell lines have been authenticated using short tandem repeat (STR) profiling.

### EV isolation from cell culture media
Cells were cultured in EV-depleted medium (by centrifugation at 100,000 g for 18 h) for 48–60 h before EV collection. The supernatants were collected from cell culture and centrifuged as a standard protocol for EV isolation[19,20]. First, cell debris and dead cells in culture supernatants were removed by centrifugation at 3,000 g for 20 min (Beckman Coulter, Allegra X-15R). Then, microvesicles were pelleted and discarded after 16,500 g centrifugation for 45 min (Beckman Coulter, Optima XE-90). Finally, the obtained supernatants were centrifuged at 100,000 g for 2 h, the pelted EVs were resuspended in PBS and collected by ultracentrifugation at 100,000 g for 2 h. The whole centrifugation operations were conducted at 4 °C.

To generate MGL EVs, 50 μM Ac$_4$ManNAz was added to the cell medium and incubated for 60 h. For MGL EV acquisition at different time points, the supernatant was collected after cell with Ac$_4$ManNAz for 0 h, 4 h, 6 h, 36 h, 60 h, respectively.

### EV characterization
For TEM (transmission electron microscope) analysis, EV samples were loaded onto copper EM grids, stained with phosphotungstic acid for 1 min, and then washed with deionized water and observed by TEM (Hitachi, ht-7700, Japan). The size distribution and zeta potential of A375 EVs were characterized by Dynamic light scattering (DLS) (Nano-ZS). Total protein level of EVs was quantified using bicinchoninic acid assay (BCA protein assay).

### General procedures of flow cytometry analysis and confocal imaging for EVs
Ten μg EVs were mixed with 4 μL aldehyde/sulfate latex beads for 15 min adsorption at room temperature. The EV-bead complexes were then blocked by 100 μL PBS with 1 M glycine and 20% BSA for 30 min. After washing twice by PBS with 0.5% BSA, the beads were pelleted by centrifugation (3,968 g, 5 min) (Eppendorf, Centrifuge 5424 R), and resuspended in 40 μL PBS with 0.5% BSA. Four μL of these EV-modified beads were incubated with anti-CD63 antibody (IgG as a control) for 1 h. After washing twice by PBS buffer with 0.5% BSA, the fluorescence intensities were measured by flow cytometry (BD FACSVerse™) and fluorescence confocal microscopy (Leica SP8-STED 3X). The acquired data of flow cytometry were analyzed by BD FACSuite Flow Cytometry software.

For MGL EVs, Ac$_4$ManNAz was added to the cell medium and incubated for 60 h. The supernatants were collected and processed as

described in the section of "EV isolation from cell culture media". Then 10 µg MGL EVs were coupled to 4 µL latex beads following the same protocol above in this section. Four µL of the MGL EV-modified beads were incubated with DBCO-Cy5 for 1 h click chemistry reaction in 100 µL PBS with 0.5% BSA. After washing twice by PBS with 0.5% BSA, the MGL EV-modified beads were analyzed by flow cytometry and fluorescence confocal microscopy. To optimize the additive quantity of Ac$_4$ManNAz, a series of different concentrations (0, 20, 50, 80 µM) of Ac$_4$ManNAz were mixed with the cells. The collected MGL EVs were reacted with 10 µM DBCO-Cy5 for the following analysis. To optimize the DBCO-Cy5 concentration, the collected MGL EVs after 50 µM Ac$_4$ManNAz treatment were reacted with a series of different concentrations of DBCO-Cy5 (0, 10, 12.5, 15 µM) for the following analysis.

## Sample preparation for proteomics

MGL EVs or non-MGL EVs from A375 cells were prepared as described in the section of "EV isolation from cell culture media". About 50 µg EVs were incubated in the freshly prepared lysis buffer (0.1 M Tris buffer with 8 M urea, pH 8.0) on ice for 20 min. Then the lysate was centrifuged at 12,000 g for 10 min at 4 °C, after which the supernatant was transferred to a new tube and the protein concentration was determined by BCA assay. The lysate was flash-frozen and stored at −80 °C until measurement by LC-MS/MS.

## Chip fabrication and modification

The herringbone chip was fabricated using standard lithography techniques and polydimethylsiloxane (PDMS) casting techniques[42,43]. The chips were made by casting the mixture of PDMS monomer and initiator (m/m = 10:1) on molds, followed by 135 °C incubation at 5 min. After solidification, PDMS slides were peeled off, punched and plasma bonded to glass coverslips.

After the fabrication, the chip was modified to the following: firstly, the PDMS chip was activated by oxygen plasma, incubated with 4% (v/v) (3-Mercaptopropyl) trimethoxysilane (MPTS) (in ethanol) for 1 h at 25 °C and then washed with ethanol before drying in a 100 °C oven for 1 h. Next, freshly prepared n-γ-maleimidobutyryl-oxysuccinimide ester (GMBS) was poured into the chip for 30 min. After washing the device with PBS, streptavidin (SA) at 20 µg mL$^{-1}$ was incubated in the device at 4 °C and storage. For anti-mouse CD63 antibody modified chip, biotinylated anti-mouse CD63 antibody at 20 µg mL$^{-1}$ was incubated in the SA modified chip for 1 h at 25 °C.

## Microfluidic detection of MGL EVs

MGL EVs were isolated as described in the section of "EV isolation from cell culture media". Five µg EVs were incubated with 12.5 µM DBCO-PEG$_4$-biotin in 20 µL PBS containing 0.5% BSA at 37 °C for 1 h. The products were loaded into the SA-Chip with a flow rate of 1.25 mL h$^{-1}$. Next, the microfluidic chip was washed with 200 µL PBS buffer with 0.5% BSA at 1.5 mL h$^{-1}$ to remove any nonspecifically bounded EVs. After incubation with SuperBlock™ Blocking buffer for 45 min, the primary antibody (anti-CD63 or anti-PD-L1) at 20 µg mL$^{-1}$ was incubated in the chip for 1 h. Then the β-galactosidase conjugated secondary antibody at 120 µg mL$^{-1}$ was incubated in the chip for 1 h. Following PBS washing, FDG (the substrate for β-galactosidase) was added. After 20 min reaction, fluorescence image was captured by fluorescence microscope (Nikon Ti-U). The fluorescence intensity was automatically calculated using ImageJ.

To establish the calibration curve for nascent PD-L1$^+$ EVs, a series of different concentrations of 4T1 or B16F10 MGL EVs (calculated by BCA assay) were incubated with 12.5 µM DBCO-PEG$_4$-biotin for 1 h reaction. The products were subjected to the SA-chip and read out by the fluorescence enzyme immunoassay via sequential introduction of anti-PD-L1 antibody, β-galactosidase conjugated secondary antibody, and FDG. For the calibration curve of the nascent CD63$^+$ EVs, same procedure was performed except that the primary antibody was replaced by the anti-CD63 antibody. For the calibration curve for the total PD-L1$^+$ CD63$^+$ EVs, the products were subjected to the anti-CD63 modified chips and read out by the fluorescence enzyme immunoassay via sequential introduction of anti-PD-L1 antibody, β-galactosidase conjugated secondary antibody, and FDG.

## Mouse models

All animal experiments were carried out in accordance with the protocol approved by Principles of Laboratory Animal Care (People's Republic of China). Female BALB/c strains and female C57BL/6 J strains of mice at 6–8 weeks of age were purchased from Xiamen University Laboratory Animal Center. All mice were housed in Animal Care Center of Xiamen University (at 20–24 °C, relative humidity of 40–60%, a 12 h light/dark cycle). Mice were weighed every 3 days. Tumors were measured using a digital caliper, and tumor volume was calculated by the formula[44]: (width)$^2$ × length × 0.52. For anti-tumor immunotherapy, the maximal tumor burden of 10% body weight was permitted by Certification and Accreditation Administration of the People's Republic of China. In all models used in this study, this limitation has not been exceeded.

For 4T1 models with immunotherapy, $2.5 \times 10^5$ 4T1 cells were subcutaneously injected into BALB/c mice, and a dose of 12.5 mg kg$^{-1}$ PD-L1 antibody or PBS buffer was injected intraperitoneally every 7 days from day 7 after implantation[36]. Simultaneously, 100 µL 140 mM Ac$_4$ManNAz was injected intraperitoneally[25] and 20 µL 25 mM was injected intratumorally[26] once daily for three consecutive days. On days 10, 17 and 24, mice were euthanized for blood.

For B16F10 models with immunotherapy, $2.5 \times 10^5$ B16F10 cells were subcutaneously injected into C57BL/6 J mice, and a dose of 12.5 mg kg$^{-1}$ PD-L1 antibody or PBS buffer was injected intraperitoneally every 2 days from day 5 after implantation. Simultaneously, 100 µL 140 mM Ac$_4$ManNAz was injected intraperitoneally[25] and 20 µL 25 mM was injected intratumorally[26] once daily for three consecutive days. On days 8, 10 and 12, mice were euthanized for blood. Blood samples were kept in EDTA K$_2$ anticoagulation tube and centrifuged 3300 g for 30 min to obtain cell-free plasma. The plasma samples were stored at -80 °C until further use.

## Isolation of tissue EVs

Tumor tissue or tissue from major organs was cut and dissociated in pieces of ~2 × 2 × 2 mm size, and then was incubated with collagenase D and DNase I at 37 °C for 30 min. Digested tissues were passed through a sterile 70 µm cell strainer on the top of a 50 mL polypropylene tube. And the filtrate was centrifuged at 3000 g. The supernatant was then collected and centrifuged at 100,000 g to isolate the tissue-derived EVs[27].

## Microfluidic detection of EVs with in vivo MGL

Five µL mouse plasma or 5 µg tissue-derived EVs was incubated with 12.5 µM DBCO-PEG$_4$-biotin in 20 µL PBS with 0.5% BSA at 37 °C for 1 h. The products were loaded into the SA-Chip with a flow rate of 1.25 mL h$^{-1}$. The subsequent process was the same as in the section of "Microfluidic detection of MGL EVs".

For biodistribution profiling, $2.5 \times 10^5$ 4T1 cells were subcutaneously injected into BALB/c mice. On day 21, Ac$_4$ManNAz was injected (20 µL of 25 mM Ac$_4$ManNAz in i. t. and 100 µL of 140 mM Ac$_4$ManNAz in i. p.) once daily for three consecutive days to 4T1 tumor-bearing mice. On day 24, the mice were euthanized for blood, tumor, heart, liver, lung, and kidney isolation. Then tissue-derived EVs were separated as described in the section of "Isolation of tissue EVs" and detected as described above in this section (using anti-CD63 as the primary antibody).

To investigate the time window for the analysis of in vivo MGL EVs with Melac-Chip, $2.5 \times 10^5$ 4T1 cells were subcutaneously injected into BALB/c mice. On day 14, Ac$_4$ManNAz was injected (20 µL of 25 mM

Ac$_4$ManNAz in i. t. and 100 μL of 140 mM Ac$_4$ManNAz in i. p.) once daily to 4T1 tumor-bearing mice for up to 4 days. The mouse plasma samples were collected each day starting from day 14, and detected as described above in this section (using anti-PD-L1 as the primary antibody).

To detect the pre-existing PD-L1$^+$ EVs, 5 μL mouse plasma was incubated with 12.5 μM DBCO-PEG$_4$-biotin in 20 μL PBS with 0.5% BSA at 37 °C for 1 h. Then, the products were incubated with 5 μL SA-beads (GE Life) at 37 °C for 1 h to remove the nascent EVs through beads capturing. After centrifugation (587 g for 5 min), the supernatant was collected and loaded to the anti-CD63 modified chip. The detection signals were then read out by the fluorescence enzyme immunoassay via sequential introduction of anti-PD-L1 antibody, β-galactosidase conjugated secondary antibody, and FDG.

### Immunofluorescence staining for tissue sections
Immunofluorescence staining was performed on paraffin-embedded (FFPE) sections. For FFPE sections, antigen retrieval by steaming in citrate buffer (pH = 6.0) was performed before blocking. The FFPE sections were incubated with primary antibodies overnight at 4 °C, followed by incubation with fluorophore-conjugated secondary antibodies for 1 h. The cell nuclei were stained with DAPI. Samples were observed using an Ortho-Fluorescent Microscopy (Pannoramic 250 FLASH).

### Statistics and reproducibility
Statistical analyses were performed using GraphPad Prism 8.3. Results are represented as mean ± standard deviation. Statistical significance was determined using paired two-tailed Student's $t$-tests or one-way analyses of variance (ANOVA) when appropriate. Asterisks are used to indicated statistical significance (*$P < 0.05$, **$P < 0.01$, ***$P < 0.001$, ****$P < 0.0001$), and n.s. indicates non-significance ($P > 0.05$). All experiments were independently repeated at least three times with similar results. No statistical method was used to predetermine sample size. No data were excluded from the analyses. The experiments were not randomized. The investigators were not blinded to allocation during experiments and outcome assessment.

### Reporting summary
Further information on research design is available in the Nature Portfolio Reporting Summary linked to this article.

## Data availability
The mass spectrometry proteomics data have been deposited to the ProteomeXchange Consortium via the PRIDE[45] partner repository with the dataset identifier PXD045742. All the data generated in this study are provided in the Supplementary Information and Source Data file. Source data are provided with this paper.

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

## Acknowledgements

This work was supported by the National Natural Science Foundation of China Grants (22022409 and 92269114 to Y.S., 22293031 to C.Y.), and the Program for Changjiang Scholars and Innovative Research Team in University (Grant IRT13036 to C.Y.) for their financial support. We thank Yaying Wu, Zheni Xu, and Dr. Changchuan Xie (School of Life Sciences, Xiamen University) for mass spectrometry experiments and data analysis.

## Author contributions

Y.S. conceived the study, designed the experiments, and revised the manuscript. Q.W. performed EV isolation, MGL for EV, chip fabrication, microfluidic detection of EVs, the animal study, and wrote the manuscript. W.W. performed the MGL for cell-derived EVs, and sample preparation for proteomics. C.Z., Z.Y., Y.Z., and S.Z. assisted the animal study. X.L. and Y.L. contributed to the numerical simulation and design of the chip. Q.W., C.Y. and Y.S. analyzed the experimental data. All authors contributed to the manuscript.

## Competing interests

The authors declare no competing interests.
