## [Peer Review File · Nature Communications]

Reviewers' Comments:

Reviewer #1:

Remarks to the Author:

This manuscript describes an in vivo labeling approach using Ac4ManNAz injection to introduce azide groups and assume this azide group could be bio-generated into EV/exosome formation and expressed on surface. Then, authors described streptavidin-modified herringbone microfluidic chip to capture such azide biotin tagged EVs. It has been an extremely difficult time to understand this manuscript, mainly due to the writing, word of choice, and grammars through out of manuscript. For example:

1. this manuscript is using cellular biogenesis process for packing azide into EV secretory pathway, the "synthesis" would not be the right choice of word to accurately describe this process.
2. Term of "metabolic pulse labeling" is very confusing

Most importantly, the rationale for this work is not clear at all. Authors totally ignored the biodistribution and bioavailability profiles on general nanoparticles, molecules, and EVs in in vivo system. The accumulation and clearance rate of injected Ac4ManNAz should be investigated first to understand how long the injected Ac4ManNAz can still be uptaken by tumor tissue cells locally to produce tagged EVs, although it is IP injection and intratumoral injection. It is actually very surprising to see EVs were still in Fluorescence after three weeks.

Additionally, Ac4ManNAz is Azide-containing Metabolic Glycoprotein Labeling Probe, which does not have any specificity to a certain cell type. Mostly, it would tag cell surface membrane from all type of cells. Thus, the efficiency that subsequently tags produced EVs from uptaken cells would expect to be low. Such details need to be careful characterized.

Although, authors claimed that temporal differentiation is the main point of this manuscript, the overall work did not reflect the ability of temporal differentiation, considering the time point is only before and after three days upon stimulation. EVs before stimulation may already exit or be cleared, and the accumulation could be too low to interfere stimulation produced EVs. Figure 4 is very confusing. Figure 4 d, e, and f indicated the lower level of PDL-1 EVs (e) than total PDL-1 EVs (f) under both treated and untreated conditions. However, figure 4 g showed higher level of PDL-1 EVs than total PDL-1 EVs under the similar conditions as figures e and f.

In figure 3 c grey bar, the non-tagged cell produced EVs also showed relatively high fluorescence detection background, which raised concerns on the detection specificity.

Figure 2 c caption is noted as the "Optimal flow rate for azide labeled exosome capture by chip", however, it did not represent or explain the two data bars in the figure.

Overall, too many technical rigors and concerns in this manuscript to prevent from considering publication.

Reviewer #2:

Remarks to the Author:

The manuscript entitled, "Temporal Differentiation of Extracellular Vesicles by Metabolic Glycan Labeling-Assisted Microfluidics" reported a novel method to measure newly synthesized exosome vesicles (EVs) from tumors and determined the clinical significance of this approach to monitor immunotherapy using 4T1 breast cancer model. Although there are some interesting findings in this study, there are some major concerns.

1. The authors should comprehensively compare the differences between newly generated EVs versus "outdated" EVs in tumor-bearing mice. In addition, many cells secrete EVs. It is unclear how to differentiate tumor-derived EVs versus EVs from other cells in this study.
2. Figure 4. The authors treated 4T-1-tumor bearing mice with anti-PD-L1 mAb and then used anti-PDL-1 mAb to capture PD-L1+ newly synthesized EVs. However, it is likely anti-PD-L1 mAb could mask/block this capture thus leading to the low yield of PD-L+ newly synthesized PD-L1+

EVs. To rule out this possibility, the authors should treat these mice with anti-PD-1 mAb and then use anti-PD-L1 mAb to capture EVs.

3. The authors should use additional tumor models, particularly immunotherapy resistant tumors such as melanoma B16-F10 to perform this experiment. This will further strengthen the overall conclusion.

Reviewer #3:

Remarks to the Author:

The manuscript from the Song lab described the use of a couple of different technologies to isolate more newly generated extracellular vesicles (EVs) from the total EVs. They accomplish this through the use of carbohydrate metabolic labeling with an azide-containing mannosamine derivative (ManNAz) originally developed by Bertozzi and coworkers. This results in the new EVs bearing azides on their associated cell-surface proteins. These azides can then be labeled with biotin using the strain-promoted azide-alkyne cycloaddition reaction. The authors then employ microfluidics to enrich and characterize the EVs using antibodies. The experiments developing the system and the author's ability to capture new EVs after fairly short timeframes are very convincing and nicely performed. The authors then move on to validate their method by looking at the amount of PDL1 in EVs in a mouse tumor model treated with anti-PDL1 therapy. Impressively, they can indeed collect EVs from these mice. I feel like this paper may be appropriate for publication in Nat Communication, but I do have one comment that should be addressed first.

Comment #1 - The difference between the total and new EVs in Figure 4e and f in terms of PDL1 amounts are subtle. I might be missing something, but it appears to me that the major difference is less PDL1 in the new EVs at week 2 in the untreated animals. Could the authors please explain their model and how they are parsing these differences? Their explanation of the new EVs being more "sensitive" to changes doesn't seem obvious to me. Alternatively, the two groups don't necessarily need to be different for the EV enrichment method to work, and may also be an interesting result.

Point-by-point response

Reviewers' comments:

Reviewer #1 (Comments for the Author):

This manuscript describes an *in vivo* labeling approach using Ac₄ManNAz injection to introduce azide groups and assume this azide group could be bio-generated into EV/exosome formation and expressed on surface. Then, authors described streptavidin-modified herringbone microfluidic chip to capture such azide biotin tagged EVs. It has been an extremely difficult time to understand this manuscript, mainly due to the writing, word of choice, and grammars through out of manuscript.

Response: We greatly appreciate the reviewer for the constructive comments and have revised the manuscript accordingly. The revision includes the addition of one panel of new data (Figure 3c-g) and nine new supplementary figures (Figure S1-3, Figure S6-11) as well as substantial text rewording. Indeed, the entire manuscript has been re-written to make it more understandable and easier-to-follow to address the reviewer's overall concerns.

For example:

1. This manuscript is using cellular biogenesis process for packing azide into EV secretory pathway, the "synthesis" would not be the right choice of word to accurately describe this process. Term of "metabolic pulse labeling" is very confusing.

Response: We thank the reviewer for pointing out this wording issue. We have changed "synthesis" to "production", and "newly synthesized" to "newly-produced" or "nascent". Also, we have replaced the sentence of

"The relative amounts of exosomes that are produced after each immunotherapy administration can then be quantitated by microfluidic separation and antibody staining, by sequential administration and corresponding metabolic pulse labeling" with

"With the Melac-Chip strategy, we were able to quantify the relative amount of EV produced after each immunotherapy using sequential administration".

In addition, we have gone through the main text and the supplementary materials to fix typos, grammar issues, and inaccurate wordings.

2. Most importantly, the rationale for this work is not clear at all. Authors totally ignored the biodistribution and bioavailability profiles on general nanoparticles, molecules, and EVs in vivo system. The accumulation and clearance rate of injected Ac₄ManNAz should be investigated first to understand how long the injected Ac₄ManNAz can still be uptaken by tumor tissue cells locally to produce tagged EVs, although it is IP injection and intratumoral injection. It is actually very surprising to see EVs were still in fluorescence after three weeks.

Response: We apologize for missing the biodistribution and bioavailability profiles in the previous version of the manuscript. In the revision, the data of biodistribution profile (revised Fig. 3c), accumulation and clearance profile (Fig. 3g), and the administration manner optimization (Fig. 3e) were presented in Figure 3 and discussed in the main text:

*“To monitor biodistribution of the azido-modified EVs, we collected the mouse samples of plasma, tumor tissue, and tissue from four major organs (heart, liver, lung and kidney). EVs in the plasma samples were directly analyzed by Melac-Chip, while EVs from tissue samples were separated by using a combination of slicing and enzyme digestion followed by differential ultracentrifugation²⁷ (Fig. 3b), and then subjected to Melac-Chip analysis. To enable the analysis, only 5 μ L of each sample was required to click with DBCO-PEG₄-biotin, and the analysis showed a similar phenomenon to that of the *in vitro* MGL, that is, the azido-labelled samples (regardless of from plasma or tissue) displayed apparent increases in the intensity of anti-CD63 staining as comparing to the PBS control (Fig. 3c), suggesting the successful MGL on EVs *in vivo* and the isolation capability of Melac-Chip on the *in vivo* labelled EV samples. Among the quantified EV levels from the *in vivo* MGL model, those from the plasma and tumor tissue samples were the highest (~43-fold higher to the lowest) (Fig. 3c), indicating that during the 3-day *in vivo* MGL window, the labelled EVs were primarily distributed within the tumor and blood.”*

C

Fig. 3c, Schematic for the detection of biotin-linked MGL CD63⁺ EVs and the detected distribution in plasma and different tissue samples. The fluorescence detection was on CD63⁺ EVs from plasma, tumor tissue and other major organs of 4T1 tumor-bearing mice with MGL, as well as CD63⁺ EVs from plasma and tumor tissue of 4T1 tumor-bearing mice without MGL. $\Delta FL = FL - FL_0$, where FL_0 and FL are the fluorescence intensity detected by Melac-Chip before and after the addition of sample. $n = 5$.

“We also studied the time window of MGL for efficient detection of these EVs with Melac-Chip. Again, we administered $Ac_4ManNAz$ (20 μ L of 25 mM $Ac_4ManNAz$ in *i. t.* and 100 μ L of 140 mM $Ac_4ManNAz$ in *i. p.*) once daily to 4T1 tumor-bearing mice for up to 4 days. We collected the mouse plasma samples each day starting from day 0. Melac-Chip analysis revealed the detectable signal of PD-L1⁺ MGL EVs on day 2 and an increase in the signal till day 3 when it seemed to be saturated (**Fig. 3g**). As the detected PD-L1⁺ MGL EV signal reached the plateau on day 3, a stop of $Ac_4ManNAz$ injection for consecutive 7 days (referring to the classical interval between two immunotherapy doses³²⁻³⁴) completely diminished the signal of this EV, suggesting the well clearance of this unnatural sugar *in vivo* and the well sensitivity of Melac-Chip.”

Fig. 3g, Normalized PD-L1 fluorescence intensity of plasma samples (n = 3) from 4T1 tumor-bearing mice at 0, 1, 2, 3 and 4 d injection of $Ac_4ManNAz$ (once/day). “3 d + & 7 d -” indicates a stop for 7 days after 3 consecutive days of injection. *** $P < 0.001$, and n.s. indicates non-significance ($P > 0.05$).

“Considering both intratumoral (*i. t.*)^{26, 30} and intraperitoneal (*i. p.*) injection^{25, 31} are commonly used for $Ac_4ManNAz$ -based cell labelling *in vivo*, we gave a series of *i. t.* and *i. p.* injection to 4T1 tumor-bearing mice (once daily for three consecutive days). Among them, using Melac-Chip we identified the optimal condition to be the combination of 20 μ L of 25 mM $Ac_4ManNAz$ in *i. t.* and 100 μ L of 140 mM $Ac_4ManNAz$ in *i. p.*, under which the anti-PD-L1 antibody staining intensity reached the highest, ~34-fold to that of the control group (PBS injection, **Fig. 3e**).”

Fig. 3e, The detected fluorescence intensity of PD-L1⁺ EVs from plasma samples of 4T1 tumor-bearing mice (n = 3) with different injection manners of Ac₄ManNAz. I. t. refers to intratumoral injection and i. p. refers to intraperitoneal injection.

Meanwhile, we have added the experimental methods for biodistribution and time window profiling in the Methods of “Microfluidic detection of EVs with *in vivo* MGL”:

“For biodistribution profiling, 2.5×10⁵ 4T1 cells were subcutaneously injected into BALB/c mice. On day 21, Ac₄ManNAz was injected (20 μL of 25 mM Ac₄ManNAz in i. t. and 100 μL of 140 mM Ac₄ManNAz in i. p.) once daily for three consecutive days to 4T1 tumor-bearing mice. On day 24, the mice were euthanized for blood, tumor, heart, liver, lung, and kidney isolation. Then tissue-derived EVs were separated as described in the section of “Isolation of tissue EVs” and detected as described above in this section (using anti-CD63 as the primary antibody).”

*“To investigate the time window for the analysis of *in vivo* MGL EVs with Melac-Chip, 2.5×10⁵ 4T1 cells were subcutaneously injected into BALB/c mice. On day 14, Ac₄ManNAz was injected (20 μL of 25 mM Ac₄ManNAz in i. t. and 100 μL of 140 mM Ac₄ManNAz in i. p.) once daily to 4T1 tumor-bearing mice for up to 4 days. The mouse plasma samples were collected each day starting from day 14, and detected as described above in this section (using anti-PD-L1 as the primary antibody).”*

As to the reviewer’s questioning of “EVs were still in fluorescence after three weeks”, we think that is misunderstanding likely caused by our previously inaccurate writing. We meant that during the weekly antibody injections, we simultaneously conducted the metabolic glycan labelling of EVs for three consecutive days. So, by using the fluorescence enzyme immunoassay the fluorescence EVs detected day 24 were actually labelled on day 18-20. We have revised the relevant description in the main text for clarity:

*“We firstly established an immunotherapy response model on 4T1-bearing mouse³³ by annotating PD-L1 antibody (PBS injection as the control) on day 7 and once/week thereafter for 4T1 breast tumor mouse (Fig. 4a). Simultaneously, we injected Ac₄ManNAz with the weekly immunotherapy for three consecutive days to ensure the saturation of *in vivo* MGL on nascent EVs according to the results of our *in vivo* MGL test (Fig. 3g).”*

3. Additionally, Ac₄ManNAz is Azide-containing Metabolic Glycoprotein Labeling Probe, which does not have any specificity to a certain cell type. Mostly, it would tag cell surface membrane from all type of cells. Thus, the efficiency that subsequently tags produced EVs from uptaken cells would expect to be low. Such details need to be careful characterized.

Response: We thank the reviewer for this comment. We agree with the reviewer that Ac₄ManNAz mostly tags N-glycosylation sites of proteins on cell membrane with azido groups. A portion of the N-glycosylation proteins on the cell membrane can transfer to the EV membrane, by outward budding of the cell surface membrane, or by an intracellular endocytic trafficking pathway involving fusion of multivesicular late endocytic compartments with the cell surface membrane (*Nature Cell Biol.* **2019**, 21, 9-17). According to the GlycoProtDB and UniProt database, 11 of 22 identified pan-EV protein markers (*Cell* **2020**, 182, 1044-1061) are N-glycosylation proteins (including HSP90AB1, CD63, CD9, CD81, FN1, LGALS3BP, A2M, JCHAIN, HBB, B2M and FLNA), suggesting the feasibility of labelling EV membrane by azido groups.

In this work, the successful incorporation of azido groups into EV surface was characterized by several methods, including flow cytometry analysis (revised Fig. 1b), confocal imaging (revised Supplementary Fig. 1), and Melac-chip analysis (Fig. 2c, d). These results all showed a multi-fold increase in the detection signal of azido-labelled EVs compared to unlabeled ones:

“Flow cytometry analysis (**Fig. 1b**) and confocal imaging (**Supplementary Fig. 1**) displayed apparent fluorescence signals from the dye on the latex beads containing MGL exosomes only (in comparison with the non-MGL control), suggesting the successful incorporation of azido groups. The detected fluorescence intensity reached the plateau at the conditions of 50 μ M Ac₄ManNAz (60-h incubation) and 12.5 μ M DBCO-Cy5 (1-h incubation).”

Fig. 1, Characterization of the azido-labelled A375 exosomes. a, Schematic diagram of the click reaction between the azido-modified A375 exosomes and the alkyl dye DBCO-Cy5. **b**, Fluorescence intensity of the A375 exosomes treated with different concentrations of Ac₄ManNAz (left) or DBCO-Cy5 (right). The exosomes were loaded on latex beads, and then reacted with the DBCO-Cy5 dye prior to characterization.

Supplementary Fig. 1. Fluorescence intensity of the MGL A375 exosomes treated with DBCO-Cy5. After treating A375 cells with 50 μM Ac₄ManNAz for 60 h, the collected MGL exosomes were loaded on latex beads, and then reacted with 12.5 μM DBCO-Cy5 (PBS as the control) for 1 h prior to characterization. MGL refers to metabolic glycan labelling.

“Using the fully developed Melac-Chip strategy, we quantified an increase of ~6.9 folds in fluorescence intensity for the MGL over the non-MGL exosomes, after removal of the exosome-null background signal (Fig. 2c, d; Supplementary Fig. 3).”

Fig. 2c, The fluorescence intensity of MGL A375 exosomes (in orange) and non-MGL A375 exosomes (in grey) captured by Melac-chip. $\Delta\text{FL} = \text{FL} - \text{FL}_0$, where FL_0 and FL are the fluorescence intensity detected by Melac-Chip before and after the addition of exosomes. **** $P < 0.0001$. $n = 3$. **d,** Representative fluorescence images of MGL A375 exosomes and non-MGL A375 exosomes captured by Melac-chip.

In addition, by choosing the detection antibody that targets a specific biomarker, we can selectively detect specific cell derived-EV subtypes, even though the Ac₄ManNAz-based MGL is not cell-selective. For example, PD-L1 is specifically and highly expressed on the surface of tumor cell-derived EVs (*Nature* **2018**, 560, 382-386; *Cell* **2019**, 177, 414-427; *Cell Res.* **2019**, 28, 862-864), and EV PD-L1 can be applied as a tumor diagnostic marker (*Nat. Rev. Immunol.* **2023**, 23, 236-250; **2020**, 20, 209-215). In the revision, we have added a few sentences and a figure (Figure 3 f) to show the selective detection of tumor-cell derived MGL EVs by Melac-Chip using the anti-PD-L1 antibody:

“As the tumor-specific antibodies are integrated into Melac-Chip for EV detection, in principle the strategy could be applied for analysis of tumor-derived EVs, even though the Ac₄ManNAz-based MGL is not tumor-selective. To demonstrate this speculation, we tried to harness the anti-PD-L1 antibody to detect the PD-L1 positive (PD-L1⁺) EVs (**Fig. 3d**), a promising marker for tumor diagnosis^{20, 28, 29}, from the total captured ones.”

“Meanwhile, at this condition nearly negligible PD-L1⁺ MGL EV signal was detected in the plasma samples from normal mice (**Fig. 3f**), supporting the tumor-biomarker role of PD-L1⁺ EV^{20, 29}.”

Fig. 3f, The detected fluorescence intensity of PD-L1⁺ EVs from plasma samples of normal mice and 4T1 tumor-bearing mice with Ac₄ManNAz treatment (n = 5).

4. Although, authors claimed that temporal differentiation is the main point of this manuscript, the overall work did not reflect the ability of temporal differentiation, considering the time point is only before and after three days upon stimulation. EVs before stimulation may already exit or be cleared, and the accumulation could be too low to interfere stimulation produced EVs.

Response: We thank the reviewer for this comment. Compared with the previous inability to isolate newly-produced EVs, our work is the first time to realize the capturing of newly-produced EVs upon stimulation. To express it more accurately, we have changed the statement of "temporal differentiation" to "isolation of newly-produced or nascent" in the revised manuscript.

As to whether the accumulated pre-existing EVs affect the detection of the stimulation-produced EVs, it depends mainly on the relative content of the two. We have selected PD-L1⁺ exosomes as the model vesicle system to compare the levels of the newly-produced and pre-existing, and added the detection results in Supplementary Fig. 8a:

Supplementary Fig. 8. Analyzing nascent and pre-existing PD-L1⁺ exosomes in response to PD-L1 immunotherapy. a. The detected concentrations of nascent PD-L1⁺ exosomes and pre-existing PD-L1⁺ exosomes at different time points.

For a more direct comparison, we calculated the percentages of the nascent PD-L1⁺ exosomes and pre-existing ones relative to the total PD-L1⁺ exosomes (shown in the table 1 below). The results revealed that the pre-existing PD-L1⁺ exosomes accounted for 37%-53% of the total PD-L1⁺ exosomes in the treated group, and 38%-65% in the untreated group. The pre-existing PD-L1⁺ exosomes occupied close to or more than half of the total ones, and were dynamically responsive to stimuli, resulting in non-negligible interference on nascent EV detection. As to this issue, Dr. Pieter Vader from University Medical Center Utrecht also emphasized the importance of developing new strategies for EV separation with time resolution and proposed that “All of these approaches have limitations in tracking the fate of EVs, and novel approaches with large dynamic ranges of both temporal and spatial resolution are required to overcome these” (*Nat. Rev. Mol. Cell Biol.* **2022**, 23, 369-382).

Days (d)	Treated		Untreated	
	Nascent PD-L1 ⁺ Exosomes	Pre-existing PD-L1 ⁺ Exosomes	Nascent PD-L1 ⁺ Exosomes	Pre-existing PD-L1 ⁺ Exosomes
10	63%	37%	35%	65%
17	50%	50%	56%	44%
24	47%	53%	62%	38%

Table 1. The proportions of nascent PD-L1 exosomes and pre-existing ones in total PD-L1 exosomes.

5. Figure 4 is very confusing. Figure 4 d, e, and f indicated the lower level of PDL-1 EVs (e) than total PDL-1 EVs (f) under both treated and untreated conditions. However, figure 4 g showed higher level of PDL-1 EVs than total PDL-1 EVs under the similar conditions as figures e and f.

Response: We apologize for the confusing in Figure 4. This is mostly caused by our data interpretation manner and wording. To make Figure 4 clearer and more accurate, we have revised Figure 4d-g in two aspects, and modified our discussion in the main text.

On one hand, to make the level of different targets more comparable, we replaced the ordinate title “*FL intensity (a. u.)*” with “*Concentration (ng μL^{-1})*”, and revised the coordinate range of Figure 4 e and f to the same. We think that the concentration comparison is more reasonable than the direct comparison of the fluorescence intensity, because the nascent EVs (d & e) were detected by the developed Melac-Chip strategy, while the total EV levels (f) were detected by a regular Chip-based method (CD63 antibody for capture).

On the other hand, we have replaced “*New Total EVs*” and “*Total PD-L1⁺ EVs*” (targets based on the detection or capturing of CD63 antibody) with “*Nascent CD63⁺ EVs*” and “*Total PD-L1⁺ Exosomes*”, respectively. Because the CD63 antibody-based recognition mainly targets exosomes not all EV subtypes, and no effective universal antibody has been reported to recognize all EV subtypes (*Cell* **2020**, 182, 1044-1061).

In the main text, we have modified the wording to discuss these results:

*“After calibration (**Supplementary Fig. 6a**), we calculated the concentrations of the newly-produced CD63⁺ EVs and found no significant difference between the treated and untreated groups (**Fig. 4d**), suggesting that the total plasma exosome production remains stable regardless of tumor growth and drug stimulation. Next, we moved to the nascent PD-L1⁺ EVs. Since the core of Melac-Chip is click chemistry-based capture followed by primary antibody recognition and secondary antibody amplification, the possible presence of anti-PD-L1 coverage, that is, certain PD-L1⁺ EVs may carry anti-PD-L1 antibody arising from the therapy, would in principle not affect the capturing (dependent on the azido-modification) nor the amplification (similar secondary-antibody-recognizable Fc regions on both the therapeutic and the detection anti-PD-L1 antibodies). To demonstrate this speculation, we extracted MGL exosomes from 4T1 cells and used them as a model to test the detection by Melac-Chip in the presence or absence of pre-incubation with the therapeutic anti-PD-L1 antibody, which showed a detectable PD-L1⁺ exosome level close to each other (**Supplementary Fig. 7**). Similar to the data processing of nascent CD63⁺ EVs, we established a calibration curve for the PD-L1⁺ EVs (**Supplementary Fig. 6b**) and then estimated that the treatment had a significant impact on these EVs’ production (**Fig. 4e**), which positively correlated to the tumor volume, with a slight decrease from day 10 to 17 and a slight increase in the week after (**Fig. 4 b, e**). As a comparison, instead of Melac-Chip we tried a regular Chip-based method (CD63 antibody for capture) that allows for the quantification of total PD-L1⁺ exosomes (a predictor for immunotherapy response²⁰) (**Supplementary Fig. 6c, Fig. 4f**). It turned out that the total PD-L1⁺ exosomes showed a good correlation with the tumor volume only in the untreated group. In the treated group, levels of these exosomes increased till day 10 and then stayed stable thereafter, which was poorly correlated with the change trend of the tumor volume during this time period (**Fig. 4b**). Further Pearson correlation analysis confirmed the good correlation between nascent PD-L1⁺ EVs and tumor volume in the treated ($R^2 = 0.9611$) and untreated ($R^2 = 0.9958$) groups (measured by Melac-Chip), as well as the good correlation between total PD-L1⁺ exosomes and tumor volume in the untreated ($R^2 = 0.8886$) but not the treated ($R^2 = 0.2663$) group (measured by a Chip-based method) (**Fig. 4g**). These data suggest that nascent*

PD-L1⁺ EVs that are only measurable by Melac-Chip can be a good indicator for tumor progression and immunotherapy response.

Fig. 4. **d-f**, Schematics of the detection of nascent CD63⁺ EVs (d), nascent PD-L1⁺ EVs (e), and total PD-L1⁺ exosomes (f) as well as the detected concentrations at different time points. ** $P < 0.01$, *** $P < 0.001$, **** $P < 0.0001$. **g**, Pearson correlation of the nascent PD-L1⁺ EVs (orange) and total PD-L1⁺ exosomes (black) to the tumor volume in 4T1-bearing mice with /without anti-PD-L1 treatment. $n = 5$ for the untreated group, $n = 6$ for the treatment group.

6. In figure 3c grey bar, the non-tagged cell produced EVs also showed relatively high fluorescence detection background, which raised concerns on the detection specificity.

Response: We thank the reviewer for this comment. To further look into the background from the EVs produced by non-tagged cells, we have used Melac-Chip to compare the detection signals of blank sample (without the addition of exosomes), exosomes from MGL untreated and treated cells (Supplementary Fig. 3). The detected fluorescence intensities were 30 (a. u.), 46 (a. u.) and 139 (a. u.), respectively. The results indicate that the fluorescence background of the non-tagged cell produced EVs is mostly caused by the inherent background of the detection system (such as the autofluorescence of FDG or the chip, tiny non-specific adsorption of antibodies), rather than due to the poor detection specificity.

Supplementary Fig. 3. Fluorescence intensity of blank sample (without the addition of exosomes), non-MGL exosomes (in grey), and MGL exosomes (in orange) captured by Melac-chip. Their fluorescence intensities were 30 (a. u.), 46 (a. u.) and 139 (a. u.), respectively. Therefore, the fluorescence background of non-tagged cell produced exosomes is mostly caused by the inherent background of the detection system (such as the autofluorescence of FDG or the chip, tiny non-specific adsorption of antibodies), rather than non-MGL exosomes. $n = 4$.

To show this more clearly, the fluorescence intensities of all samples in Figure 3c and other relevant results (Figure 2c) were re-calculated by removing of the exosome-null background signals (Supplementary Fig. 3). Meanwhile, we have also modified the corresponding contents in the main text:

“Using the fully developed Melac-Chip strategy, we quantified an increase of ~6.9 folds in fluorescence intensity for the MGL over the non-MGL exosomes, after removal of the exosome-null background signal (Fig. 2c, d; Supplementary Fig. 3).”

Fig. 2c, The fluorescence intensity of MGL A375 exosomes (in orange) and non-MGL A375 exosomes (in grey) captured by Melac-chip. $\Delta FL = FL - FL_0$, where FL_0 and FL are the fluorescence intensity detected by Melac-Chip before and after the addition of exosomes. **** $P < 0.0001$. $n = 3$.

“To enable the analysis, only 5 μ L of each sample was required to click with DBCO-PEG₄-biotin, and the analysis showed a similar phenomenon to that of the *in vitro* MGL, that is, the azido-labelled samples (regardless of from plasma or tissue) displayed apparent increases in the intensity of anti-CD63 staining as comparing to the PBS control (Fig. 3c), suggesting the successful MGL on EVs *in vivo* and the isolation capability of Melac-Chip on the *in vivo* labelled EV samples.”

c

Fig. 3c, Schematic for the detection of biotin-linked MGL CD63⁺ EVs and the detected distribution in plasma and different tissue samples. The fluorescence detection was on CD63⁺ EVs from plasma, tumor tissue and other major organs of 4T1 tumor-bearing mice with MGL, as well as CD63⁺ EVs from plasma and tumor tissue of 4T1 tumor-bearing mice without MGL. $\Delta FL = FL - FL_0$, where FL_0 and FL are the fluorescence intensity detected by Melac-Chip before and after the addition of sample. $n = 5$.

7. Figure 2c caption is noted as the “Optimal flow rate for azide labeled exosome capture by chip”, however, it did not represent or explain the two data bars in the figure.

Response: We thank the reviewer for pointing out this issue. We have revised Figure 2c caption to:

“The fluorescence intensity of MGL A375 exosomes (in orange) and non-MGL A375 exosomes (in grey) captured by Melac-chip. $\Delta FL = FL - FL_0$, where FL_0 and FL are the fluorescence intensity detected by Melac-Chip before and after the addition of exosomes”.

Meanwhile, we have added a sentence and a supplementary figure to show the results of flow rate optimization:

“To isolate the MGL exosomes, a biorthogonal reaction was performed by using alkyl biotin (12.5 μM DBCO-PEG₄-biotin for 1 h) to click with the exosomal azido group, the products of which were later captured by streptavidin (SA) modified microchip (**Fig. 2a**) with an optimized flow rate of 1.25 mL h^{-1} (**Supplementary Fig. 2**) (unless otherwise specified, these conditions were used in the subsequent analysis by Melac-Chip).”

Supplementary Fig. 2. The intensity ratio of the MGL exosomes treated with Ac₄ManNAz versus untreated at different flow rates in Melac-Chip. n = 3.

8. Overall, too many technical rigors and concerns in this manuscript to prevent from considering publication.

Response: We thank the reviewer for the critiques. We have carefully revised the whole manuscript to make three substantial improvements. Firstly, a series of data were added to comprehensively and systematically establish the Melac-Chip strategy, including the condition optimization of *vitro* MGL (revised Figure 2b), the optimization of microfluidic flow rate (Supplementary Fig. 2), the distribution (revised Figure 3c) and metabolic time of azido modified EVs *in vivo* (Figure 3g), the optimization of administration manner (Figure 3e), selective characterization (revised Figure 3c, Supplementary Fig. 1 & 3), the effect of therapeutic PD-L1 antibody on MGL exosome detection (Supplementary Fig. 7), and additional tumor model (Supplementary Fig. 9-11). Secondly, we have revised the introduction to further clarify the innovations, emphasizing the importance of selectively sorting nascent EVs from the total, which was not possible before this work. Thirdly, we have improved the readability of the main text and the supplementary materials by fixing typos, grammatical issues, and inaccurate wording. Overall, we think that our revision has addressed all concerns from this reviewer and the revised manuscript is now suitable for publication.

Reviewer #2 (Remarks to the Author):

The manuscript entitled, "Temporal Differentiation of Extracellular Vesicles by Metabolic Glycan Labeling-Assisted Microfluidics" reported a novel method to measure newly synthesized exosome vesicles (EVs) from tumors and determined the clinical significance of this approach to monitor immunotherapy using 4T1 breast cancer model. Although there are some interesting findings in this study, there are some major concerns.

Response: We thank the reviewer for the supportive and constructive comments.

1. The authors should comprehensively compare the differences between newly generated EVs versus "outdated" EVs in tumor-bearing mice. In addition, many cells secrete EVs. It is unclear how to differentiate tumor-derived EVs versus EVs from other cells in this study.

Response: Thanks for the suggestion! Since "outdated" EV capture depends on antibody recognition and no effective antibody has been reported to recognize all EV subtypes (*Cell* **2020**, 182, 1044-1061), we selected exosome (which can be captured by CD63 antibody) as the model vesicle for comparison. The results were presented as a supplementary figure (Supplementary Fig. 8) and discussed in the main text:

*"Besides, we were also curious about the change trends of the nascent and pre-existing PD-L1⁺ exosomes. After using the azido-induced isolation to remove the nascent EVs instead of detecting them, we were able to capture the pre-existing PD-L1⁺ exosomes by anti-CD63 antibody and then detect them by anti-PD-L1 antibody (See Methods "Microfluidic detection of EVs with in vivo MGL" for details). The levels of nascent PD-L1⁺ exosomes were generated via the subtraction of the pre-existing ones from the total that were measured in **Fig. 4f**. By plotting these data together (**Supplementary Fig. 8**), we found that in the untreated group both the nascent ($R^2 = 0.8914$) and pre-existing ($R^2 = 0.8547$) PD-L1⁺ exosomes exhibited the relatively good correlation to tumor volume, but in the treated group neither of them correlated well ($R^2 = 0.2424$ & 0.1963 for the former and latter, respectively). Comparing to the high correlation between nascent PD-L1⁺ EV and tumor volume, the poor correlation between nascent PD-L1⁺ exosomes and tumor volume hints that certain subtypes of EVs other than exosome are more directly responsible to the immunotherapy."*

Supplementary Fig. 8 Analyzing of nascent and pre-existing PD-L1⁺ exosomes in response to PD-L1 immunotherapy. **a**, The detected concentrations of nascent PD-L1⁺ exosomes and pre-existing PD-L1⁺ exosomes at different time points. **b**, Pearson correlation of the nascent PD-L1⁺ exosomes (orange) and pre-existing PD-L1⁺ exosomes (purple) to the tumor volume in 4T1-bearing mice with /without anti-PD-L1 treatment. $n = 5$ for the untreated group, $n = 6$ for the anti-PD-L1 treatment group.

For the detection selectivity, although all cells secrete EVs, our method can achieve selective detection of tumor-derived EVs by using the detection antibody specifically targeting tumor biomarker. In our work, we applied anti-PD-L1 antibody as detection antibody for tumor-derived EV detection (Figures 4e and f), as PD-L1 is known to be specifically and highly expressed on the surface of tumor cell-derived EVs (*Nature* **2018**, 560, 382-386; *Cell* **2019**, 177, 414-427; *Cell Res.* **2019**, 28, 862-864). We have added two sentences and a figure (Fig. 3f) to further explain this in the main text:

“As the tumor-specific antibodies are integrated into Melac-Chip for EV detection, in principle the strategy could be applied for analysis of tumor-derived EVs, even though the Ac_4ManNAz -based MGL is not tumor-selective. To demonstrate this speculation, we tried to harness the anti-PD-L1 antibody to detect the PD-L1 positive (PD-L1⁺) EVs (Fig. 3d), a promising marker for tumor diagnosis^{20, 28, 29}, from the total captured ones.”

“Meanwhile, at this condition nearly negligible PD-L1⁺ MGL EV signal was detected in the plasma samples from normal mice (Fig. 3f), supporting the tumor-biomarker role of PD-L1⁺ EV^{20, 29}.”

Fig. 3f, The detected fluorescence intensity of PD-L1⁺ EVs from plasma samples of normal mice and 4T1 tumor-bearing mice with Ac₄ManNAz treatment (n = 5). $\Delta FL = FL - FL_0$, where FL₀ and FL are the fluorescence intensity detected by Melac-Chip before and after the addition of sample.

2. Figure 4. The authors treated 4T-1-tumor bearing mice with anti-PD-L1 mAb and then used anti-PDL-1 mAb to capture PD-L1⁺ newly synthesized EVs. However, it is likely anti-PD-L1 mAb could mask/block this capture thus leading to the low yield of PD-L⁺ newly synthesized PD-L1⁺ EVs. To rule out this possibility, the authors should treat these mice with anti-PD-1 mAb and then use anti-PD-L1 mAb to capture EVs.

Response: In Figure 4e, PD-L1⁺ newly-produced EVs were captured by the interaction between biotin on the EVs (dependent on the azido-modification) and streptavidin on the chip, and the detection signal was obtained by sequential introduction of anti-PD-L1 antibody, β -galactosidase-coupled rabbit anti-IgG H and L, and FDG. Whether EVs were covered by therapeutic PD-L1 antibody or not does not affect the capture. As for the signal readout, since both the therapeutic and detection anti-PD-L1 isotypes were IgG, which can be identified by the used secondary antibody, the potential PD-L1 antibody coverage on EVs would not affect the subsequent binding of the secondary antibody. We have added two sentences and a supplementary figure to discuss the effect of therapeutic PD-L1 antibody-induced coverage on nascent EV detection:

“Next, we moved to the nascent PD-L1⁺ EVs. Since the core of Melac-Chip is click chemistry-based capture followed by primary antibody recognition and secondary antibody amplification, the possible presence of anti-PD-L1 coverage, that is, certain PD-L1⁺ EVs may carry anti-PD-L1 antibody arising from the therapy, would in principle not affect the capturing (dependent on the azido-modification) nor the amplification (similar secondary-antibody-recognizable Fc regions on both the therapeutic and the detection anti-PD-L1 antibodies). To demonstrate this speculation, we extracted MGL exosomes from 4T1 cells and used them as a model to test the detection by Melac-Chip in the presence or absence of pre-incubation with the therapeutic anti-PD-L1 antibody, which showed a

detectable PD-L1⁺ exosome level close to each other (**Supplementary Fig. 7**).”

Supplementary Fig. 7 Schematic of the detection of PD-L1 positive MGL-exosomes, as well as the detected intensity of MGL-exosomes with anti-PD-L1 blockade (in blue) and without blockade (in grey). Because both therapeutic and detectable anti-PD-L1 isotypes were rat IgG, which can be identified by secondary antibody (β-galactosidase-coupled rabbit anti-rat IgG H and L). $\Delta FL = FL - FL_0$, where FL_0 and FL are the fluorescence intensity detected by Melac-Chip before and after the addition of exosomes. $n = 5$. n. s. indicates non-significance ($P > 0.05$).

3. The authors should use additional tumor models, particularly immunotherapy resistant tumors such as melanoma B16-F10 to perform this experiment. This will further strengthen the overall conclusion.

Response: Thanks for the suggestion! We have added B16F10 mouse model to evaluate the generality of Melac-Chip:

“To show the generality of Melac-Chip, we switched to the B16F10 melanoma-bearing mouse model which is resistant to the immune-checkpoint blockade³⁸ (**Supplementary Fig. 9, 11 a-c**) for the quantification of nascent PD-L1⁺ EVs. After calibration (**Supplementary Fig. 10**), the detected levels of nascent PD-L1⁺ EVs again exhibited a good correlation to tumor volume in both the untreated and treated group ($R^2 = 0.9741$ vs 0.9902 , **Supplementary Fig. 11 d, g**). Meanwhile, regular chip analysis also indicated the good correlation between pre-existing PD-L1⁺ exosomes and tumor volume ($R^2 = 0.9528$ vs 0.9586) as well as total PD-L1⁺ exosomes and tumor volume ($R^2 = 0.9712$ vs 0.9220) in the two groups (**Supplementary Fig. 11 e-g**). However, among the three indexing vesicles, nascent PD-L1⁺ EVs still correlated best to tumor volume, regardless of immunotherapy or not, which coincides well with the results in the 4T1 breast cancer mouse model. Therefore, Melac-Chip enables detection of nascent PD-L1⁺ EVs in both immunotherapy responded

and resistant models, providing a timely and accurate reflection of tumor progression and therapy effect.”

Supplementary Fig. 11 EV analysis for B16F10-bearing mouse model with immunotherapy. **a**, Schematic of the tumor implantation, PD-L1 immunotherapy, metabolic glycan labelling and sample collection in a B16F10-bearing mouse model. **b**, Tumor growth curves of B16F10-bearing mice with/without PD-L1 antibody treatment. **c**, Representative images of tumor tissue by HE and immunofluorescent staining. HE staining for general histology, TUNEL assay to identify and quantify apoptotic cells, CD8⁺ T cells to measure tumor infiltration, and Ki-67 protein for tumor cell proliferation. **d-f**, Schematic of the detection of nascent PD-L1⁺ EVs

(d), pre-existing PD-L1⁺ exosomes (e), and total PD-L1⁺ exosomes (f) as well as the detected concentrations at different time points. * $P < 0.05$, *** $P < 0.001$, and n.s. indicates non-significant ($P > 0.05$). **g**, Pearson correlation of the nascent PD-L1⁺ EVs (orange), pre-existing PD-L1⁺ exosomes (purple) and total PD-L1⁺ exosomes (black) to the tumor volume in B16F10-bearing mice with/without anti-PD-L1 treatment. $n = 5$.

Reviewer #3 (Remarks to the Author):

The manuscript from the Song lab described the use of a couple of different technologies to isolate more newly generated extracellular vehicles (EVs) from the total EVs. They accomplish this through the use of carbohydrate metabolic labeling with an azide-containing mannosamine derivative (ManNAz) originally developed by Bertozzi and coworkers. This results in the new EVs bearing azides on their associated cell-surface proteins. These azides can then be labeled with biotin using the strain-promoted azide-alkyne cycloaddition reaction. The authors then employ microfluidics to enrich and characterize the EVs using antibodies. The experiments developing the system and the author's ability to capture new EVs after fairly short timeframes are very convincing and nicely performed. The authors then move on to validate their method by looking at the amount of PD-L1 in EVs in a mouse tumor model treated with anti-PD-L1 therapy. Impressively, they can indeed collect EVs from these mice. I feel like this paper may be appropriate for publication in Nat Communication, but I do have one comment that should be addressed first.

Response: We thank the reviewer for the supportive and encouraging comments.

1. The difference between the total and new EVs in Figure 4e and f in terms of PD-L1 amounts are subtle. I might be missing something, but it appears to me that the major difference is less PD-L1 in the new EVs at week 2 in the untreated animals. Could the authors please explain their model and how they are parsing these differences?

Response: Thanks for the question! We apologize for the confusing in Figure 4. This is mostly caused by our data interpretation manner and wording. To make Figure 4 clearer and more accurate, we have revised Figure 4e-f in two aspects, and modified our discussion in the main text.

On one hand, we replaced the ordinate title “*FL intensity (a. u.)*” with “*Concentration (ng μL^{-1})*”, and revised the coordinate range of Figure 4 e and f to the same. We think that the concentration comparison is more reasonable than the direct comparison of the fluorescence intensity, because the nascent EVs (e) were detected by the developed Melac-Chip strategy, while the total EV levels (f) were detected by a regular Chip-based method (CD63 antibody for capture).

On the other hand, we have replaced “*Total PD-L1⁺ EVs*” (the target based on the capturing of CD63 antibody) with “*Total PD-L1⁺ Exosomes*”. Because the CD63 antibody-based recognition mainly targets exosomes not all EV subtypes, and no effective universal antibody has been reported to recognize all EV subtypes (*Cell* **2020**, 182, 1044-1061). It is

not appropriate to compare the two levels with each other, as one is of exosome (Figure 4e) and the other is of EV (Figure 4f).

In the main text, we have modified the wording to discuss these results:

“Similar to the data processing of nascent CD63⁺ EVs, we established a calibration curve for the PD-L1⁺ EVs (**Supplementary Fig. 6b**) and then estimated that the treatment had a significant impact on these EVs’ production (**Fig. 4e**), which positively correlated to the tumor volume, with a slight decrease from day 10 to 17 and a slight increase in the week after (**Fig. 4 b, e**). As a comparison, instead of Melac-Chip we tried a regular Chip-based method (CD63 antibody for capture) that allows for the quantification of total PD-L1⁺ exosomes (a predictor for immunotherapy response²⁰) (**Supplementary Fig. 6c, Fig. 4f**). It turned out that the total PD-L1⁺ exosomes showed a good correlation with the tumor volume only in the untreated group. In the treated group, levels of these exosomes increased till day 10 and then stayed stable thereafter, which was poorly correlated with the change trend of the tumor volume during this time period (**Fig. 4b**).”

Fig. 4. e-f, Schematics of the detection of nascent PD-L1⁺ EVs (e) and total PD-L1⁺ exosomes (f) as well as the detected concentrations at different time points. ** $P < 0.01$, *** $P < 0.001$, **** $P < 0.0001$.

For the major difference between nascent PD-L1⁺ EVs (4e) and total PD-L1⁺ exosomes (4f) at week 2 in the untreated animals, it is possible that the tumor is relatively small and less active in the second week (4b), so the recent secretion is less than the accumulation of the pre-secretion PD-L1⁺ vesicles (which is known to correlate with tumor progression. *Nature* **2018**, 560, 382–386; *Cell* **2019**, 177, 414–427; *Cell Res.* **2019**, 28, 862–864). By the third week, the tumor gets larger and more active. As a result, the total PD-L1⁺ exosomes are mainly from the recently secreted ones, so the difference between the two types of vesicles is smaller than week 2.

2. Their explanation of the new EVs being more "sensitive" to changes doesn't seem obvious to me. Alternatively, the two groups don't necessarily need to be different for the EV enrichment method to work, and may also be an interesting result.

Response: We conclude that new PD-L1⁺ EV is more sensitive to change of tumor volume, because in the 4T1-bearing mouse of the anti-PD-L1 immunotherapy model, the positive

correlation of nascent PD-L1⁺ EV level to tumor volume ($R^2 = 0.9611$) is much better than that of total PD-L1⁺ exosome ($R^2 = 0.2663$, Figure 4g). To make it clearer, we have modified Figure 4g and the corresponding wording in the main text:

“Further Pearson correlation analysis confirmed the good correlation between nascent PD-L1⁺ EVs and tumor volume in the treated ($R^2 = 0.9611$) and untreated ($R^2 = 0.9958$) groups (measured by Melac-Chip), as well as the good correlation between total PD-L1⁺ exosomes and tumor volume in the untreated ($R^2 = 0.8886$) but not the treated ($R^2 = 0.2663$) group (measured by a Chip-based method) (Fig. 4g). These data suggest that nascent PD-L1⁺ EVs that are only measurable by Melac-Chip can be a good indicator for tumor progression and immunotherapy response.”

Fig. 4g, Pearson correlation of the nascent PD-L1⁺ EVs (orange) and total PD-L1⁺ exosomes (black) to the tumor volume in 4T1-bearing mice with /without anti-PD-L1 treatment. n = 5 for the untreated group, n = 6 for the treatment group.

To quantify these two groups (nascent PD-L1⁺ EV (Fig. 4e) and total PD-L1⁺ exosome (Fig. 4f)), different capture methods are still required at present. The enrichment of the nascent PD-L1⁺ EV is dependent on the azido-modification based Melac-Chip, while total PD-L1⁺ exosome enrichment is based on a CD63 antibody-modified chip, which is a regular method for exosome isolation¹⁻². Because our aim is to compare the respective correlations between these two types of vesicles and tumor size rather than the differences in their content, the different enrichment methods should not affect the results of the correlation coefficients.

Ref

1. Lim, C.Z.J., Zhang, Y., Chen, Y. *et al.* Subtyping of circulating exosome-bound amyloid β reflects brain plaque deposition. *Nat Commun* **10**, 1144 (2019).
2. Zhao, W., Hu, J., Liu, J. *et al.* Si nanowire Bio-FET for electrical and label-free detection of cancer cell-derived exosomes. *Microsyst Nanoeng* **8**, 57 (2022).

Reviewers' Comments:

Reviewer #1:

Remarks to the Author:

Although authors made substantial revision to add more data, but it raised more concerns meantime:

1. From figure 3f, It is surprising to see PD-L1+ EVs from normal mice plasma sample is closed to zero. It is common to see PD-L1 is not expressed from normal tissue sample but increased in tumor tissue. However, in plasma sample with majority of immune cells, PD-L1 is expressed mainly on T cells, B cells, DCs, and macrophages and is further up-regulated upon activation (Regulation and Function of the PD-L1 Checkpoint, Immunity 48, March 20, 2018). Thus, this data is controversial with general recognition.

2. From two model validations in both Figure 4 d, f and Figure s11 d, f, the total PD-L1+ EVs can correlate very well with tumor volume as the nascent PD-L1+ EVs. So it is not convincing on the idea to have complicated labeling process for monitoring nascent PD-L1+ EVs.

3. Using the term of "exosomes" will need to characterize by origin of endosome pathway, if not, extracellular vesicles would be a better term to reduce the confusion. In this manuscript, some places are using exosomes and some are using EVs which needs to keep a consistency.

Reviewer #2:

Remarks to the Author:

The authors have satisfactorily addressed the concerns raised in my previous review.

Reviewer #3:

Remarks to the Author:

The authors have addressed my comment sufficiently. I support publication of the manuscript.

Point-by-point response

Reviewers' comments:

Reviewer #1 (Comments for the Author):

Although authors made substantial revision to add more data, but it raised more concerns meantime:

1. From figure 3f, It is surprising to see PD-L1⁺ EVs from normal mice plasma sample is closed to zero. It is common to see PD-L1 is not expressed from normal tissue sample but increased in tumor tissue. However, in plasma sample with majority of immune cells, PD-L1 is expressed mainly on T cells, B cells, DCs, and macrophages and is further up-regulated upon activation (Regulation and Function of the PD-L1 Checkpoint, Immunity 48, March 20, 2018). Thus, this data is controversial with general recognition.

Response: We thank the reviewer for raising up this issue! Like the reviewer pointed out, some types of immune cells are known to express PD-L1. However, similar to our results, PD-L1 has been reported to be practically undetectable on EVs/exosomes in normal mouse/human plasma samples by using classical protein detection methods. For example, Professor Yoshitaka Nagai from Osaka University applied mass spectrometry analysis to normal mouse blood EVs and failed to identify the PD-L1 protein (*PLoS One* 2022,17(6), e0270634, S1 Table). Professor Wei Guo from University of Pennsylvania found that the ELISA detected levels of PD-L1 exosomes from healthy plasma samples were close to zero (*Nature* 2018, 560, 382, Extended Data Fig. 3a; shown as Figure 1a here). Using nano-flow cytometry, Professor Xiaomei Yan from Xiamen University found that the concentration of PD-L1 EVs in normal donor plasma samples was significantly lower than that in patients with nasopharyngeal carcinoma (*Anal. Chem.* 2022, 94(27), 9740, Fig. 3a; shown as Figure 1b here).

Figure 1 Reported PD-L1 exosome/PD-L1 EV levels in healthy donor plasma samples. (a) ELISA showing the level of PD-L1 on circulating exosomes in plasma samples from healthy donors (HD, n = 11) and melanoma patients (MP, n = 44). The exosomes were purified using differential centrifugation (refer to *Nature* 2018, 560, 382, Extended Data Fig. 3a). (b) Plasma concentrations measured for EV subsets exposing each of the five protein markers (LMP1, LMP2A, PD-L1, EGFR, and EpCAM) for nasopharyngeal carcinoma patients (n = 42, mean \pm s.e.m. (standard error of the mean)) and healthy donors (n = 40, mean \pm s.e.m.) by nano-flow

cytometry (refer to *Anal. Chem.* 2022, 94, 9740, Fig. 3a).

We suspect that the situation of the theoretically existed but practically undetectable PD-L1 EVs may be due to the extremely low level of PD-L1⁺ EVs in normal mouse plasma. In the future, by developing more sensitive PD-L1 detection methods, we should be able to answer this question better.

In the revised manuscript, we have cited the reference that was pointed out by this reviewer (Regulation and Function of the PD-L1 Checkpoint, *Immunity* 48, March 20, 2018), and modified our words a bit in the main text to explain this. We have replaced the sentence of

“Meanwhile, at this condition nearly negligible PD-L1⁺ MGL EV signal was detected in the plasma samples from normal mice (Fig. 3f), supporting the tumor-biomarker role of PD-L1⁺ EV^{20, 29}.” with

“Meanwhile, although PD-L1 is known to be expressed by immune cells in the plasma³², at this condition nearly negligible PD-L1⁺ MGL EV signal was detected in the plasma samples from normal mice (Fig. 3f), suggesting low expression of PD-L1 on EVs in normal plasma samples^{20,33,34} and supporting the tumor-biomarker role of PD-L1⁺ EV^{20, 29}”.

Ref.20 Chen, G. *et al.* Exosomal PD-L1 contributes to immunosuppression and is associated with anti-PD-1 response. *Nature* **560**, 382-386 (2018).

Ref.32 Sun, C., Mezzadra R. & Schumacher, T. N. Regulation and function of the PD-L1 checkpoint. *Immunity* **48**, 434-452 (2018)

Ref.33 Xiao, M. Z., Takeuchi, T., Takeda, A., Mochizuki, H. & Nagai, Y. Comparison of serum and plasma as a source of blood extracellular vesicles: Increased levels of platelet-derived particles in serum extracellular vesicle fractions alter content profiles from plasma extracellular vesicle fractions. *PLoS One* **17**, e0270634 (2022)

Ref.34 Yun, Y. H. *et al.* Noninvasive diagnosis of nasopharyngeal carcinoma based on phenotypic profiling of viral and tumor markers on plasma extracellular vesicles. *Anal. Chem.* **94**, 9740-9749 (2022)

2. From two model validations in both Figure 4 d, f and Figure s11 d, f, the total PD-L1⁺ EVs can correlate very well with tumor volume as the nascent PD-L1⁺ EVs. So it is not convincing on the idea to have complicated labeling process for monitoring nascent PD-L1⁺ EVs.

Response: Thanks for bringing up this issue! As this reviewer noticed, the total PD-L1⁺ EVs can correlate well with tumor volume. But, the well correlation only occurs in the non-treatment (Figure 4f, 4g left) and immunotherapy-resistant models (Figure s11 d, f, g). In the immunotherapy-responsive model, the correlation ($R^2 = 0.2663$) of total PD-L1⁺ EVs to

tumor volume is much poorer than that ($R^2 = 0.9611$) of nascent PD-L1⁺ EVs to tumor volume (Figure 4g right). Thus, nascent PD-L1⁺ EVs has the potential to be a more accurate predictor than total PD-L1⁺ EVs for all kinds of tumor progression and therapy response. To further clarify that, we have changed “total PD-L1⁺ exosomes” to “total PD-L1⁺ CD63⁺ EVs” (Please refer to the response to comment #3) and replaced the sentence of

“These data suggest that nascent PD-L1⁺ EVs that are only measurable by Melac-Chip can be a good indicator for tumor progression and immunotherapy response.” with

“These data suggest that nascent PD-L1⁺ EVs that are only measurable by Melac-Chip can serve as a more universal indicator for tumor progression and immunotherapy response than total PD-L1⁺ CD63⁺ EVs, a previously used indicator²⁰.”

Figure 4g (right) Pearson correlation of the nascent PD-L1⁺ EVs (orange) and the total PD-L1⁺ CD63⁺ EVs (black) to the tumor volume in 4T1-bearing mice with anti-PD-L1 treatment. n = 5 for the untreated group, n = 6 for the treatment group.

In addition, we think that the labelling process is not complicated. The labelling material Ac₄ManNAz can be easily obtained and the labelling operation, i.e., intratumoral and intraperitoneal injection, is routine operation that has/can be acquired by most of the chemical biology labs. Thus in our opinion, Melac-chip is a strategy easy to get started.

Moreover, the aim of this project is to develop a strategy to allow us to be able to capture nascent EVs stimulated by an external cue. As to this aspect, Dr. Pieter Vader from University Medical Center Utrecht has emphasized the importance of developing new strategies for EV separation with time resolution and proposed that “All of these approaches have limitations in tracking the fate of EVs, and novel approaches with large dynamic ranges of both temporal and spatial resolution are required to overcome these.” (*Nat. Rev. Mol. Cell Biol.* 2022, 23, 369-382). In this work, we chose the immunotherapy models to demonstrate the feasibility of Melac-Chip. In principle, the applications of Melac-Chip can be further extended to the detection of nascent EVs stimulated by other non-immunotherapy cues, such as microbial infection, environmental changes (temperature, light, and gravity), diet changes, etc. We have added a sentence in the end of Conclusion to discuss that:

“Besides adding a temporal dimension to our understanding of EV dynamics in immunotherapy, in principle Melac-Chip can also be applied to study EV dynamics that is stimulated by a broad range of cues, such as microbial infection, environmental (temperature, light, and gravity) variation, and diet change.”

Taking all above into consideration, we hope that we can convince this reviewer that to develop a strategy such as Melac-Chip is necessary and has unique value.

3. Using the term of “exosomes” will need to characterize by origin of endosome pathway, if not, extracellular vesicles would be a better term to reduce the confusion. In this manuscript, some places are using exosomes and some are using EVs which needs to keep a consistency.

Response: Thanks for the suggestion! We have taken this reviewer’s suggestion, replaced the “exosomes” with “EVs” and modified the corresponding figures.

For example, we have changed “*we selected exosome, an important subtype of EV, as the model vesicle system to test...*” to “*we selected EV isolated from cell culture media as the model vesicle system to test...*”.

Also, previously we used “*total PD-L1⁺ exosomes*” to represent EVs that are captured by CD63 antibody and detected by PD-L1 antibody. In the revision, we have modified those words to “*total PD-L1⁺ CD63⁺ EVs*”.

Reviewer #2 (Remarks to the Author):

The authors have satisfactorily addressed the concerns raised in my previous review.

Response: We thank the reviewer for the supportive comment.

Reviewer #3 (Remarks to the Author):

The authors have addressed my comment sufficiently. I support publication of the manuscript.

Response: We thank the reviewer for the supportive comment.